# `RAT` 🐨 : Bridging RNN Efficiency and Attention Accuracy via Chunk-based Sequence Modeling

**Xiuying Wei**[1]*, **Anunay Yadav**[1], **Razvan Pascanu**[2], **Caglar Gulcehre**[1]
[1]CLAIRE, EPFL [2]Google DeepMind

## Abstract

Transformers have become the cornerstone of modern large-scale language models, but their reliance on softmax attention poses a computational bottleneck at both training and inference. Recurrent models offer high efficiency, but compressing the full sequence into a fixed-size and holistic representation can suffer from memory degradation in long contexts and limit fine-grained retrieval. To address this, we propose `RAT`, an intermediate design that bridges the efficiency of RNNs and capacity of attention. `RAT` partitions the input into chunks, applies recurrence within each chunk for local dependencies, and softmax-based attention across chunks for long-range interactions. This design mitigates memory degradation and enables direct access to distant tokens, while retaining computational efficiency. Empirically, with a chunk size of 16, the `RAT` block achieves a $7\times$ improvement in training speed for 100K sequence length and $9\times$ in generation at the 4K position, while maintaining similar performance compared to standard attention. We demonstrate this by training 1.3B parameter models from scratch and performing large-scale evaluations, including short- and long-context benchmarks, as well as supervised fine-tuning (SFT). We further propose a hybrid architecture that interleaves `RAT` with local attention. By combining efficient long-range modeling with strong local interactions, this hybrid design not only improves inference speed and reduces cache memory usage, but also consistently enhances performance and shows the overall best results. Code is available at `https://github.com/CLAIRE-Labo/RAT`.

## 1 Introduction

Language modeling has long been dominated by Transformer-based architectures due to their strong performance across a wide range of tasks. However, their reliance on full self-attention [1] results in quadratic time and memory complexity with respect to sequence length, which limits scalability in long-context processing. This limitation has motivated a wave of recent efforts to revisit recurrent models or propose novel linear recurrent models such as state space models, linear attention methods [2–8].

By comparing these architectures, we observe a key difference between recurrent models and self-attention. Recurrent approaches compress the full sequence history into fixed-size and holistic representations, which can lead to degraded memory when modeling long sequences and limit precise information retrieval. In contrast, self-attention retains full-token access and thus do not suffer from the two problems, but at the cost of heavy computation. This motivates us to explore an intermediate design that partially compresses the sequence while still maintaining global access.

We propose a `RAT` layer, a simple yet effective temporal mixing method with a chunk-based design. It divides long sequences into chunks, applying recurrence within each chunk for local modeling and softmax-based attention across chunks for direct access to distant information (see Fig. 1a). Recurrence efficiently captures short-range dependencies while avoiding the memory degradation common

---

*Correspondence to `xiuying.wei@epfl.ch`.

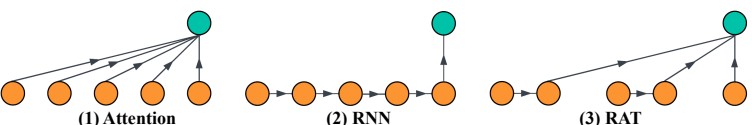
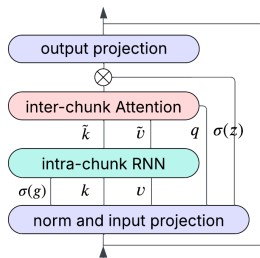

(a) **Comparison of three structures**: Circles denote token representations and arrows show the information flow for the $5^{th}$ token. (1) Attention: Each token keeps its own representation and has full access to all previous tokens. (2) RNN: Information is progressively compressed along the sequence, and the current token accesses only the last hidden state. (3) `RAT(L=2)`: Intra-chunk recurrence compresses local information, while inter-chunk attention enables direct access to previous chunks. (1) and (2) can be viewed as `RAT` with $L=1$ and $L=T$, respectively.

(b) `RAT` **structure.** The symbol $\sigma(\cdot)$ further applies a sigmoid function to the gates.

in long sequences, whereas attention over chunk-level representations enables long-range retrieval. By adjusting the chunk size $L$, `RAT` interpolates between attention (when $L=1$) and RNN (when $L=T$).

To make `RAT` scalable and efficient, we further explore its positional encoding, parameter allocation, and efficient implementations that specifically address the causal masking problem in this chunk-based design. Moreover, we investigate a hybrid architecture that interleaves `RAT` with sliding-window attention [9–12], which focuses computation on local windows. As `RAT` suggests that overusing attention in local contexts underutilizes its strengths, and such dependencies can be handled more efficiently via lightweight recurrence, the two approaches can be complementary.

We demonstrate that `RAT` is both efficient and performant. For instance, with a chunk size of $L=16$, the latency of a single temporal mixing block at position 4096 is up to $9\times$ lower than that of attention. As shown in Table 1, the full model achieves up to $10\times$ higher maximum throughput, and interleaving with local attention yields about $4\times$ improvement. We pretrain models at 1.3B scale and compare their results on 1) zero-shot commonsense reasoning with short contexts, (2) long-context understanding on LongBench [13], and (3) supervised fine-tuning on long-context tasks. `RAT` with $L=16$ performs on par with full attention in most benchmarks and even outperforms it on several LongBench tasks. When interleaved with local attention, it achieves the best overall results across all variants while maintaining high efficiency. Our main contributions are as follows:

1. We propose the `RAT` layer, a novel intermediate architecture that bridges the efficiency of recurrence and the capacity of attention. It compresses only local context while preserving global access, enabling direct retrieval and avoiding the memory degradation caused by full sequence compression in long contexts.
2. `RAT` is simple, scalable, and efficient. It requires no custom CUDA kernels and is naturally compatible with existing multi-dimension parallelism schemes. We also introduce a hybrid variant by interleaving `RAT` with sliding-window attention, enabling efficient long-range modeling with strong local interactions.
3. We validate `RAT` through extensive experiments at the 1.3B scale across diverse tasks, including 7 short-context reasoning benchmarks, 11 long-context tasks, 4 supervised fine-tuning objectives, and 9 retrieval-heavy synthetic evaluations. `RAT` with $L=16$ shows comparable performance to full attention with $9\times$ faster single-layer decoding. Its hybrid variant with local attention yields the best overall results, along with a $3-4\times$ maximum throughput boost—for instance, +1 accuracy on commonsense reasoning, +4 on code completion, +4 on a challenging QA task, and +1 on a difficult summarization task.

Table 1: Representative results for 1.3B models across pretraining, direct evaluation, and SFT. -SWA denotes interleaving with sliding-window attention (SWA) (window size 1024). Maximum throughput is measured by generating 1024 tokens given a prompt of 3072 tokens on a H100 GPU in GH200 system. See Sec. 4 for details.

| Model | Throughput | Pretrain | Direct Evaluation | | | | SFT | |
|---|---|---|---|---|---|---|---|---|
| | token/sec | Val. PPL | CSR Avg. acc | SQA Avg. F1 | Summ Avg. Rouge-L | Code Avg. EditSum | NQA[1] F1 | QMSum Rouge-L |
| Attention | 3052 | 7.61 | 56.9 | 18.2 | 19.5 | 23.9 | 61.3 | 23.4 |
| `RAT(L=16)` | 31170 | 7.67 | 56.7 | **19.6** | **20.2** | 17.4 | 60.8 | 23.3 |
| Attention-SWA | 4605 | 7.61 | 57.1 | 17.4 | 19.4 | 21.7 | **63.3** | 23.4 |
| `RAT(L=16)-SWA` | 13582 | **7.57** | **58.0** | 18.8 | 19.5 | **28.2** | 63.2 | **24.6** |

## 2 Overall architecture

To motivate our design, we first examine how attention and recurrence compress contextual information at the token level, along with the equations that will be reused later. We then introduce RAT as an intermediate mechanism that inherits the respective advantages of both.

### 2.1 Attention: Full-Token Access

In softmax attention mechanisms, each token has access to all preceding tokens through a learned token-specific weighted aggregation:

$$y_t = f(q_t K_{:}^{\top}) V_{:} \tag{1}$$

$f(\cdot)$ denotes causal masking and the softmax function, and $f(q_t K_{:}^{\top})$ represents the attention weights used to aggregate $V_{:}$. We adopt Python-style indexing in $K_{:}^{\top}, V_{:}$ to emphasize that each query attends to all keys and values. This full-token access makes attention a dominant architecture in sequence modeling, but it suffers from high computational cost in both training and inference.

### 2.2 Recurrence: Full-Sequence Compression

Recurrent models maintain a summary of past information in a fixed-size representation, allowing each step to depend only on the previous state and the current input [14, 15]. To initiate the design of RAT, we adopt a simple and fast linear recurrence [16–18], which uses diagonal matrices and no nonlinearities to simplify the classic recurrence into an EMA-like gating mechanism:

$$\begin{aligned} \tilde{v}_t &= g_t \odot \tilde{v}_{t-1} + (1 - g_t) \odot v_t, \\ y_t &= z_t \odot \tilde{v}_t, \end{aligned} \tag{2}$$

where $g_t$ and $z_t \in \mathbb{R}^D$ denote the per-dimension forget and output gates, respectively, computed via linear projections of the input followed by a sigmoid activation. Here, $D$ refers to the dimension of the model. This can be efficiently implemented using the parallel scan algorithm [18]. We take this minimal choice to highlight the core idea, but the recurrence in RAT is not limited to this certain form and can be extended to more advanced variants such as 2D recurrence or nonlinear RNNs. We leave this as future work.

The state-based formulation of recurrence yields low computational cost and efficient inference, and performs well on short sequences. However, compressing the entire sequence history into a fixed-size and holistic state, even with more expressive designs, can still suffer from memory degradation when the sequence length grows [19], and limits precise information retrieval, particularly in noisy contexts.

### 2.3 RAT: Chunk-Based Intermediate Design

To bridge the strong performance of attention enabled by full-token access with the efficiency of RNNs derived from full-sequence compression, we propose an intermediate design by reinterpreting the input as a sequence of shorter chunks. A recurrent module is applied within each chunk to model local dependencies, followed by cross-chunk attention to enable global interactions, as illustrated in Fig. 1a. This design mitigates the fixed-size representation limitation of RNNs and the inefficiency of attention.

Technically, we divide a sequence of length $T$ into $C$ chunks of length $L$, such that $T = C \cdot L$. A token originally at position $t$ is re-indexed as $(c, l)$, where $c$ denotes the chunk index and $l$ the position within the chunk. Within each chunk, a forget gate $g_{c,l}$ is used to recurrently aggregate the value $v_{c,l}$ and key $k_{c,l}$ vectors, yielding updated representations $\tilde{v}_{c,l}$ and $\tilde{k}_{c,l}$. For each query $q_{c,l}$, we compute attention on the chunk-level key and value vectors, including $\tilde{K}_{:,-1}$ for all preceding chunks, and $\tilde{k}_{c,l}$ for the current chunk. The causal masking function $f(\cdot)$ restricts attention to the chunks before it, followed by a softmax operation. Finally, an output gate is applied to produce the output:

$$\begin{aligned} \tilde{v}_{c,l} &= g_{c,l} \odot \tilde{v}_{c,l-1} + (1 - g_{c,l}) \odot v_{c,l} && \text{(Intra-chunk RNN)} \\ \tilde{k}_{c,l} &= g_{c,l} \odot \tilde{k}_{c,l-1} + (1 - g_{c,l}) \odot k_{c,l} && \text{(Intra-chunk RNN)} \\ y_{c,l} &= f([q_{c,l} \tilde{K}_{:,-1}^{\top}; q_{c,l} \tilde{k}_{c,l}^{\top}])[\tilde{V}_{:,-1}; \tilde{v}_{c,l}] && \text{(Inter-chunk attention)} \\ y_{c,l} &= z_{c,l} \odot y_{c,l} \end{aligned} \tag{3}$$

The overall computation flow is illustrated in Fig. 1b. Thanks to the short length of each chunk, the recurrent module efficiently captures local dependencies without significant information loss. Cross-chunk attention then enables global access without compressing distant information. This design preserves the accuracy of softmax-based attention while reducing computation, as inter-chunk attention operates over shorter sequences with $L$ as the FLOPs reduction ratio. We also experimented with a reversed variant by applying attention within chunks and recurrence across them, but found that standard `RAT` achieves better FLOPs utilization and overall performance (see Appendix C).

**Benefits of the chunk design**   We position this as an **intermediate architecture** between RNNs and attention, and refer to it as `RAT`. By adjusting the chunk size $L$, `RAT` interpolates between their behaviors: when $L = T$, it reduces to a pure RNN; when $L = 1$, it resembles full attention. This allows us to pursue the trade-off between attention and RNN with a single-layer design, and offers greater flexibility for hybrid modeling by using different chunk sizes to emulate varying behaviors. From a mechanistic perspective, unlike either classic or advanced recurrence (state space or linear attention models) that rely on fixed-size and holistic representations, our chunk-based structure enables memory capacity to scale with sequence length while maintaining a fixed FLOPs reduction ratio. Its partial compression with direct access to prior chunks ensures its superior performance on retrieval-heavy tasks.

## 3   Scalable and Efficient Modeling with `RAT`

### 3.1   Design details

**Parameter allocation**   We aim to control the parameters of `RAT` at $4D^2$, given $D^2$ for the output projection, $3D^2$ for the query, key, and value projections in attention, and an additional $2D^2$ for the two gates in RNN. We explore lightweight alternatives by using low-rank projections for the gates $\boldsymbol{g}$ and $\boldsymbol{z}$, or by sharing query and key projections across heads. Empirical results in Sec. 4.2 show that sharing the query $\boldsymbol{q}$ and key $\boldsymbol{k}$ slightly outperforms using low-rank gates. Notably, this design does not collapse into single-head attention, since the forget gate operates at the per-dimension level and produces distinct gated keys $\tilde{\boldsymbol{k}}$ for the inter-chunk attention.

**Positional encoding**   We examine how to encode positional information in `RAT`, due to the presence of cross-chunk attention. Motivated by the fact that RNN captures positional information through its sequential structure, we find that applying positional encoding at the chunk level, rather than relying on original token positions, yields slightly better fidelity. This strategy also improves length generalization, as the number of positions requiring encoding (i.e., the number of chunks) is much smaller than the full sequence length. In our main experiments, we use RoPE [20] based on chunk indices for inter-chunk attention. In the length generalization study (see Subsec. C.2), we further explore NoPE [21], which yields the best overall generalization performance.

**Hybrid design with Local Attention**   `RAT` is a hierarchical architecture, but not a hybrid model, as it applies the same strategy across all tokens, layers, and heads. It is compatible with various hybrid strategies and offers more flexibilities in hybrid modeling by varying the chunk size in different layers or heads. In particular, we explore interleaving `RAT` with sliding-window attention (SWA) [9–11], a widely adopted technique in recent models [22, 23, 5, 7]. We find that the two are highly complementary: while local attention methods allocate most computation within fixed windows, `RAT` reserves attention for global access and handles local modeling more efficiently. Interleaving them enables the model to efficiently and effectively capture both short-range and long-range dependencies.

### 3.2   Efficiency

We discuss efficiency-related aspects of `RAT`, and show that our current implementation does not rely on custom CUDA or Triton kernels, yet achieves significant speed-ups in both training and generation in experiments. The pseudocode for training is provided in Listing 1, and the decoding algorithm is shown in Listing 2.

To begin, the FLOPs per token of `RAT` are $\mathcal{O}(C \cdot D)$, compared to $\mathcal{O}(D)$ for RNN and $\mathcal{O}(T \cdot D)$ for full attention, where $C$ is the number of chunks, $D$ the model dimension, and $T$ the sequence length. Most components of `RAT` are simple and easy to implement, except that the chunk-based design introduces a non-trivial causal masking challenge during training, which we address below.

**Causal masking problem** In training, where tokens are processed in parallel, special care must be taken to apply causal masking in inter-chunk attention. First, a block-wise causal mask is required. Second, each token must also attend to its own chunk's key and value, which should be gated up to its position. They vary across tokens due to causal masking, preventing efficient parallel computation. To address this, we adopt an online softmax formulation [24]: we separately compute $f(\boldsymbol{q}_{c,l}\tilde{\boldsymbol{K}}_{:,-1}^{\top})\tilde{\boldsymbol{V}}_{:,-1}$ and $f(\boldsymbol{q}_{c,l}\tilde{\boldsymbol{k}}_{c,l}^{\top})\tilde{\boldsymbol{v}}_{c,l}$, and then combine the results by adjusting the softmax denominator. The first term can be implemented in parallel using existing attention frameworks, while the second is handled via a simple einsum.

**Practical implementation** For training, we implement intra-chunk recurrence in Eq. (3) using PyTorch's *associative scan*, enabling forward and backward passes with $\mathcal{O}(T)$ FLOPs. Compared to full-sequence recurrence, chunking reduces scan depth from $\mathcal{O}(\log T)$ to $\mathcal{O}(\log L)$ and thus improves parallelism. For inter-chunk attention, we use PyTorch's *flex attention* to implement the first term above. It supports *flash attention* with flexible features such as custom masks and returning the softmax denominator, and thus aligns well with our needs. For decoding, tokens are generated sequentially. Intra-chunk recurrence only requires single-step updates and can be implemented directly. For inter-chunk attention, standard implementations like *flash attention* [25] can be used without modification, as no complex causal masking is required at inference time.

**Parallelism** We think `RAT` is compatible with both tensor parallelism and context parallelism, which are commonly used for large model dimensions and long sequence lengths, respectively. Both the recurrence and cross-chunk attention in `RAT` are head-independent, making it easy to apply standard tensor parallelism by assigning different heads to different GPUs. For context parallelism, the intra-chunk recurrence is chunk-independent, allowing chunks to be distributed across GPUs. Since `RAT` stores much fewer number of chunk-level key/value vectors (e.g., 16× fewer than full attention), the cross-chunk attention may even avoid ring-style communication.

## 4 Experiments

In this section, we present the efficiency and large-scale evaluations of `RAT`, along with comparisons to other models. Additional discussions are provided in Appendix C.

### 4.1 Setup

For brevity, we summarize the setup for the 1.3B model with a 4K context window, which is used in most of our experiments. Full implementation details are available in Appendix A.

**Model** We adopt a Transformer architecture that interleaves a token mixing block with a hidden-state mixing block (FFN), each wrapped with residual connections and LayerNorm. We compare variants that use different token mixing modules, including RNN, attention, and `RAT`, as well as recent state space and linear attention models. In addition, we provide hybrid models that interleave `RAT` with sliding-window attention (window size 1024), such as Attention-SWA and `RAT-SWA`.

**Efficiency** We benchmark the latency of a single token mixing block, including input and output projections, on a single H100 GPU (GH200 system, 120GB), and also report the maximum throughput of the full model. We measure the time required to train on a full sequence or to generate a batch of tokens at specific positions. For fair comparison, we use *flash attention* for the attention baseline and *associative scan* for RNN. All models are compiled using *torch.compile* and evaluated in bfloat16 with *torch.cuda.amp*.

**Accuracy** We pretrain all 1.3B models on 100B tokens from FineWeb-Edu [26] using the same setup: a learning rate of 8.0e-4 decayed to 1.0e-6 (cosine schedule) and a global batch size of 2M tokens, following DeepSeek's hyperparameter guidelines [27]. We directly evaluate the models on both short-context tasks—several classical commonsense reasoning benchmarks [28]—and long-context tasks, including 11 tasks from LongBench [13] covering QA, summarization, and code completion. Since LongBench includes instruction-heavy prompts that pretrained models often struggle with, we also include SFT-based evaluations. Specifically, we use NarrativeQA [29] (two modes), QMSum [30], and WikiSum [31] to test long-context understanding with SFT. To assess retrieval capabilities, we

include the Ruler benchmark [32] and test nine synthetic needle-in-haystack tasks with varying configurations. A single round of lightweight fine-tuning is applied to adapt models to specific prompts.

## 4.2 Analyses of `RAT`

Figure 2: **Latency of the temporal mixing block** (including linear projections) with a model dimension of 2048. (a): full-sequence latency with 200K tokens; (b): generation of 512 tokens at specified positions. We adopt *flash attention* for Attention.

Table 2: **Maximum throughput of full models** (tokens/sec), measured by generating 1024 tokens from a 3072-token prompt. By reducing the KV cache memory and boosting speed, we achieve 10× maximum throughput compared to *flash attention*, and even more on 13B models, as attention suffers from poor GPU utilization at larger scale.

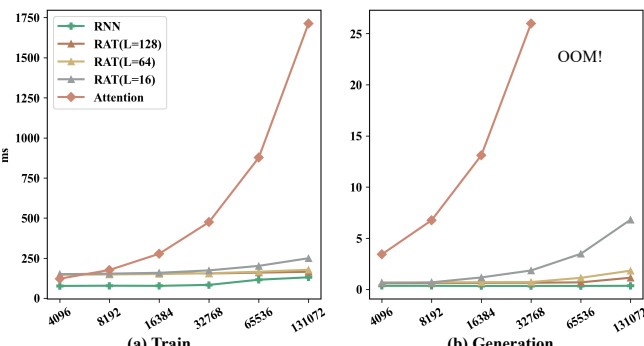

(a) Train      (b) Generation

| Model | 1.3B | 7B | 13B |
|---|---|---|---|
| RAT(L=16) | 31170 | 10103 | 5749 |
| Attention | 3152 | 983 | 534 |
| Ratio | 10.2× | 10.3× | 10.8× |

**Efficiency study** Fig. 2 and Table 2 present the efficiency comparison, with additional results provided in Appendix B. For training, on the strong H100 GPU, when the sequence length is short (e.g., 4096), `RAT` is slightly slower than attention due to underutilized GPU parallelism in the *flex attention* (with only 256 chunks) and the overhead introduced by *associative scan*. We expect these can be improved through further kernel-level optimizations. As the sequence length increases, `RAT` becomes increasingly efficient: for $L=16$, we observe approximately $2\times$ speedup at 16K, $3\times$ at 32K, $4\times$ at 64K, and $7\times$ at 100K tokens. For generation, `RAT(L=16)` achieves a $9\times$ speedup at position 4K and around $10\times$ for longer sequences. It also reduces KV cache usage, making it significantly less prone to out-of-memory (OOM) errors and enabling much higher maximum throughput.

**Ablation study** We conduct the ablation study on a 200M-parameter model trained on the book dataset. Allocating more parameters to the intra-chunk RNN gates yields significantly better performance than assigning them to the inter-chunk attention query and key projections, improving the perplexity by 0.4–0.5. Further replacing the original RoPE with a cross-chunk variant, where positions are indexed by chunk index, brings additional improvement, especially at long sequence lengths, with a 0.3 drop in PPL observed.

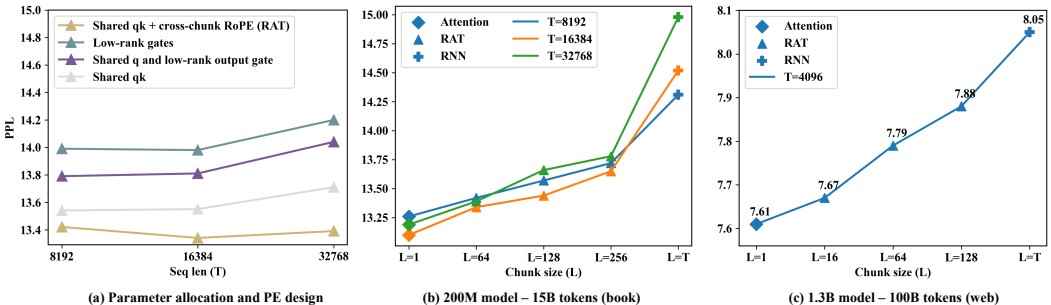

(a) Parameter allocation and PE design    (b) 200M model – 15B tokens (book)    (c) 1.3B model – 100B tokens (web)

Figure 3: (a) Ablation study on `RAT(L=64)`. (b) and (c) show pretraining results on 200M and 1.3B models, respectively. `RAT` lies between RNN and attention in terms of pretraining perplexity.

**Pretraining** In Fig. 3(b) and (c), we begin our study by examining pretraining perplexity as a foundational measure of architectural performance, demonstrating that `RAT` falls between attention and RNN. In (b), using a 200M model, we observe that increasing the chunk size leads to higher

perplexity while reducing FLOPs. Viewing attention as $L = 1$ and RNN as $L = T$, we observe perplexities of 13.26 for attention, 13.42 for `RAT(L=64)`, 13.72 for `RAT(L=256)`, and 14.31 for RNN. Interestingly, increasing the training context from $T = 8192$ to $T = 32768$ leads to a sharp rise in perplexity for RNN, while both attention and `RAT` remain stable, demonstrating the pitfalls of full-sequence compression. We then scale up to a 1.3B model trained on web data, which primarily consists of short sequences (often under 1K tokens) concatenated with separator tokens to construct 4K-length contexts. We find that `RAT(L=16)` gives the comparable performance to attention, and again `RAT` with different chunk sizes exhibits behavior intermediate between attention and RNN.

## 4.3 Large-scale evaluation and comparisons

In this section, we present direct evaluation and SFT results of our trained 1.3B models. We mainly compare `RAT(L=16)` with its two extremes: RNN and attention. Additionally, we include results from recent state space and linear attention models [5], which are trained under the same settings (model size, sequence length, and dataset) as ours. We report the value/slot sizes, which indicates the size of memory slots and FLOPs, based on sequence length $T$, model dimension $D$, and number of layers $N$. Note that state space and linear attention models use fixed-size memory and expand the classic recurrence state, while `RAT` scales its memory capacity with the sequence length.

**Commonsense reasoning**   In Table 3, the performance gap (within two points on average) between RNN and attention is small on these short commonsense reasoning tasks, indicating that RNN can be surprisingly competitive in short-context settings. `RAT(L=16)` consistently outperforms RNN on most tasks and surpasses other recent models on 4 out of 7 benchmarks, with only at most $20ND$ memory slots. We also observe that some binary-choice tasks (e.g., BoolQ, WinoGrande) show inconsistent results under small-scale pretraining. For instance, attention struggles on BoolQ, while `RAT` under-performs on WinoGrande. Interestingly, the hybrid variant `RAT(L=16)-SWA`, which interleaves `RAT` with SWA, achieves the best overall accuracy across almost all tasks. This suggests a complementary architecture: using SWA for strong local attention and `RAT` for long-range dependencies can bring us a model that is not only more efficient but also more accurate than its dense counterpart.

Table 3: We evaluate our models with model dimension $D = 2048$ and number of layers $N = 24$ on seven commonsense reasoning tasks using *lm-evaluation-harness* [28]. Since all tasks (except BoolQ, which has only 13 sequences $> 300$ tokens) have sequences $\leq 300$ tokens, we set $T = 300$. Results marked with * are from a recent study [5] trained on the same setting as ours. For SWA, slot sizes remain $150ND$ due to the short context.

| Model | Value/Slot size $T = 300$ | ARC-C acc/acc_n | ARC-E acc/acc_n | HellaSwag acc/acc_n | LAMBADA acc | PIQA acc/acc_n | WinoGrande acc | BoolQ acc |
|---|---|---|---|---|---|---|---|---|
| RNN | $ND$ | 35.15/38.57 | 70.66/66.46 | 42.43/55.95 | 39.74 | 71.49/72.69 | 53.99 | 62.54 |
| Mamba* | $64ND$ | -/35.40 | 69.52/- | -/52.91 | -/43.98 | 71.32/- | 52.95 | 61.13 |
| Mamba2* | $256ND$ | -/37.88 | 72.47/- | -/55.67 | -/45.66 | 71.87/- | 55.24 | 60.13 |
| DeltaNet* | $128ND$ | -/35.66 | 68.47/- | -/50.93 | -/42.46 | 70.72/- | 53.35 | 55.29 |
| GatedDeltaNet* | $288ND$ | -/38.39 | 71.21/- | -/55.76 | -/46.65 | 72.25/- | 57.45 | 60.24 |
| Attention | $300ND$ | 35.67/37.71 | 71.25/66.67 | **44.16/57.44** | **47.84** | **73.29/72.91** | **57.70** | 58.23 |
| RAT(L=16) | $\sim 20ND$ | **36.35/39.33** | **72.64/67.80** | 43.27/56.44 | 44.40 | 72.03/72.69 | 53.20 | **63.30** |
| GatedDeltaNet-SWA* | $288N/2D - 300N/2D$ | -/40.10 | 71.75/- | -/56.53 | 47.73 | 72.57/- | **58.4** | 63.21 |
| Attention-SWA | $300N/2D - 300N/2D$ | 35.24/37.20 | 71.97/66.20 | 44.36/57.09 | 47.97 | 72.25/72.74 | 56.91 | 61.25 |
| RAT(L=16)-SWA | $\sim 20N/2D - 300N/2D$ | **37.20/40.78** | **72.56/68.22** | **44.33/57.85** | **49.33** | **73.29/73.94** | 56.91 | **63.21** |

**LongBench**   In Table 4, we present results on 11 long-context tasks from LongBench. Since LongBench includes instruction-heavy prompts across diverse domains, it's natural that no single model dominates all tasks. Evaluating purely pretrained models on such instruction-based datasets can be very challenging. Therefore, we focus on general performance trends, especially average scores by task category. As shown in the table, `RAT(L=16)` and its hybrid variant `RAT(L=16)-SWA` achieve top performance in many groups, such as question answering and summarization. This is noteworthy, given that `RAT(L=16)` slightly lags behind attention in pretraining perplexity and underperforms in short-context commonsense reasoning. We hypothesize that its advantage on LongBench stems from structural adaptation to very long sequences—for instance, achieving 24.9 on GovReport (3-1) (vs. 18.6 for the attention baseline) and 16.7 on HotpotQA (2-1) (vs. 13.4). Second, unlike the small gap ($\sim$2 points) between RNN and attention on short-context tasks, the difference becomes much larger on LongBench, further demonstrating the problem of full-sequence compression. We also notice that

Table 4: Evaluation results on LongBench with $T = 4096$, $D = 2048$, and $N = 24$. NQA: NarrativeQA, MQA: MultiFieldQA-en, HQA: HotpotQA, WQA: 2WikiMultihopQA, MSQ: Musique, GR: GovReport, MN: MultiNews, TQA: TriviaQA, RBP: RepoBench-P. Metrics: F1 for QA tasks, Rouge-L for summarization, and EditSum for code tasks. Results marked with * are from a recent study [5] trained under the same settings as ours. Note that they use local attention with a window size of 2048, while we use 1024.

| Model | Value/Slot size | Single-Document QA | | | | Multi-Document QA | | | | Summarization | | | | Code Completion | | |
|---|---|---|---|---|---|---|---|---|---|---|---|---|---|---|---|---|
| | $T = 4096$ | NQA | Qasper | MQA | Avg. | HQA | WQA | MSQ | Avg. | GR | QMSum | MN | Avg. | LCC | RBP | Avg. |
| RNN | $ND$ | 11.7 | 8.8 | 19.5 | 13.3 | 9.1 | 15.1 | 5.4 | 9.9 | 16.9 | 16.8 | 16.4 | 16.7 | 14.0 | 17.6 | 15.8 |
| Mamba2* | $256ND$ | 11.1 | 11.3 | 18.6 | 13.7 | 11.8 | 15.1 | **6.7** | 11.2 | 6.7 | 14.5 | 7.4 | 9.5 | 17.9 | 20.6 | 19.3 |
| DeltaNet* | $128ND$ | 12.9 | 10.8 | 21.5 | 15.1 | 10.9 | 13.2 | 5.1 | 9.7 | 6.5 | 13.5 | 7.2 | 9.1 | 17.6 | 20.3 | 19.0 |
| GatedDeltaNet* | $288ND$ | 14.1 | 14.0 | 23.3 | 17.1 | 13.7 | 14.4 | 5.8 | 11.3 | 7.5 | 16.4 | 7.9 | 10.6 | 18.7 | 22.1 | 20.4 |
| Attention | $4096ND$ | 12.3 | 14.0 | 28.2 | 18.2 | 13.4 | 17.7 | 4.9 | 12.0 | 18.6 | **18.9** | 21.0 | 19.5 | **21.3** | **26.5** | **23.9** |
| RAT(L=64) | $64ND$ | 13.9 | 13.0 | 27.0 | 18.0 | 14.9 | 16.3 | 4.3 | 11.8 | 18.9 | 17.7 | 19.9 | 18.8 | 16.2 | 19.7 | 18.0 |
| RAT(L=16) | $256ND$ | **14.5** | **16.1** | **28.3** | **19.6** | **16.7** | **18.9** | 6.3 | **14.0** | **24.9** | 17.5 | 18.3 | **20.2** | 14.2 | 20.6 | 17.4 |
| GatedDeltaNet-SWA* | $288N/2D-2048N/2D$ | **14.5** | 12.3 | 26.6 | 17.8 | 12.6 | **23.6** | 6.1 | **14.1** | 9.1 | 16.1 | 12.8 | 12.7 | 15.5 | 19.2 | 17.4 |
| Attention-SWA | $4096N/2D-1024N/2D$ | 13.1 | 14.6 | 24.5 | 17.4 | 13.7 | 19.0 | 6.1 | 12.9 | 19.4 | 17.3 | **21.4** | 19.4 | 18.7 | 24.6 | 21.7 |
| RAT(L=16)-SWA | $256N/2D-1024N/2D$ | 12.7 | **15.6** | **28.2** | **18.8** | 14.2 | 18.1 | **7.4** | 13.2 | **20.1** | 18.6 | 19.8 | **19.5** | **26.3** | **30.1** | **28.2** |

RAT(L=16) performs less favorably on code completion. However, its hybrid variant significantly improves performance in this setting, outperforming the second-best model by about four points.

**Supervised Fine-tuning**    Since LongBench tasks are instruction-based and pretrained-only models often struggle to follow specific prompts, we further assess the performance of SFT on some long-context QA and summarization tasks, as shown in Table 5. For NarrativeQA, where passages are extremely long and may not contain the answer within the truncated 4K-token context, we evaluate two settings: (1) summary-only, which uses only the passage summary, and (2) summary plus passage, which concatenates the summary with the full passage. In both cases, RAT(L=16)-SWA delivers strong performance while remaining efficient. Although both the summary and passage contain useful information, we observe that models, particularly the attention-based ones, perform worse in the second setting, with F1 dropping to 33. We hypothesize that in this setup, attention may be more easily distracted due to the absence of structural bias, which can lead to flattened attention score distributions [33]. In contrast, models based on RAT appear more robust, likely due to their chunk-wise structure. For summarization tasks, RNN performs better than in QA, and RAT(L=16)-SWA achieves the best result on QMSum, surpassing the next-best model by approximately 1 Rouge-L point.

Table 5: **Left**: Average and 95th percentile of context lengths of datasets with the LLaMA2 tokenizer. **Right**: SFT performance on long-context datasets with $T = 4096$, $D = 2048$, and $N = 24$. F1 is used for QA tasks, and Rouge-L is used for summarization tasks. NarrativeQA[1]: only the summary of a passage is provided. NarrativeQA[2]: the summary plus the full passage is provided; the passage typically contains much irrelevant information.

| Task | Avg. | 95th pctl. |
|---|---|---|
| NarrativeQA[1] | 838 | 1296 |
| NarrativeQA[2] | 97k | 254k |
| QMSum | 16k | 29k |
| WikiSum | 1828 | 3473 |

| Model | Value/Slot size | NarrativeQA[1] | NarrativeQA[2] | QMSum | WikiSum |
|---|---|---|---|---|---|
| RNN | $ND$ | 30.5 | 27.4 | 23.2 | 34.3 |
| RAT(L=16) | $256ND$ | 60.8 | _43.5_ | 23.3 | 36.6 |
| Attention | $4096ND$ | 61.3 | 33.0 | _23.4_ | **36.8** |
| Attention-SWA | $4096N/2D-1024N/2D$ | **63.3** | 39.6 | _23.4_ | _36.7_ |
| RAT(L=16)-SWA | $256N/2D-1024N/2D$ | _63.2_ | **43.7** | **24.6** | _36.7_ |

Table 6: **Retrieval ability**: Accuracy performance with exact match scoring on the Needle-in-Haystack tasks with different configurations from the RULER benchmark [32]. We use $T = 4096$, $D = 2048$, and $N = 24$.

| Model | Value/Slot size | single_1 | single_2 | single_3 | multikey_1 | multikey_2 | multikey_3 | multiquery | multivalue |
|---|---|---|---|---|---|---|---|---|---|
| RNN | $ND$ | 99.4 | 99.8 | 0.0 | 2.2 | 0.0 | 0.0 | 5.8 | 5.7 |
| RAT(L=64) | $64ND$ | 100.0 | 99.8 | 55.6 | 96.4 | 62.2 | 1.0 | 86.3 | 89.1 |
| RAT(L=16) | $256ND$ | 99.6 | 100.0 | 94.6 | 99.6 | 99.6 | 82.6 | 91.2 | 94.8 |
| Attention | $4096ND$ | 100.0 | 100.0 | 100.0 | 99.6 | 99.6 | 95.2 | 98.6 | 99.1 |

**Retrieval ability**    In Table 6, we evaluate the three models on challenging synthetic needle-in-haystack tasks that test retrieval ability (see Subsec. C.3 for task descriptions). It can be seen that attention consistently performs well due to its full-token access. RNN handles simpler tasks like single_1 and single_2 reasonably well, but its performance drops on harder settings. RAT(L=16) matches attention closely on most tasks but struggles with UUID ones (single_3, multikey_3) due to their difficulty and evaluation metric as exact match scoring. Meanwhile, RAT(L=64) falls

between RNNs and `RAT(L=16)`, as expected given its partial access to long-range context. These results align with the model structures: attention sees all tokens directly, recurrence compresses all history into a single state, and `RAT` blends both, preserving chunk-level access while reducing FLOPs.

## 5 Related work

**State space and linear attention models** Recurrent neural networks (RNNs) have long been used for sequential modeling [34–36]. Recent state space models [37, 18, 2, 7, 8] extend recurrence by expanding hidden states from vectors to matrices for higher accuracy, removing non-linearity and using structured operations (e.g., diagonal matrices) to enable parallel computation [18]. Per-head gating connects these models [3] to linear attention [4, 38, 39, 8, 40, 5] and shows the duality between linear recurrence and linear attention. Unlike this duality perspective, we take an intermediate view between per-dimension-gated RNNs and softmax-based attention, without enforcing strict linearity. Linear recurrence in `RAT` is intended primarily for training efficiency, rather than imposed as a design constraint.

Despite state expansion, these models inherit the fixed-size representation limitation of classic RNNs, leading to degraded memory on long sequences. A concurrent work [41] proposes addressing this by growing the memory slots logarithmically. We similarly target the fixed-size memory problem, but use softmax-based attention to access distant tokens. Hybrid strategies [23, 42] combining state space models and attention are also popular. While `RAT` is not a hybrid with the same computation graph for all tokens and all layers, it is orthogonal to such designs and provides more flexibility, such as using different chunk sizes for different layers and heads, which we leave for future work.

**Softmax-based attention** Attention mechanisms suffer from slow computation due to their full-token access. Earlier works have explored local attention with chunking, combined with recurrence or sparse mechanisms to access earlier context [9, 10, 43–45]. For example, Ma et al. [43] gates query/key vectors before local attention, while Hua et al. [10] adds the outputs of local attention to cross-chunk linear attention. In our work, we instead leverage attention's strength in directly accessing distant tokens with lightweight recurrence for local contexts, and hierarchically organize their outputs. As shown in Sec. C.1, `RAT` utilizes FLOPs more effectively than local attention. We also regard `RAT` and local attention as structurally complementary. Although we experimented with interleaving them in this work, we leave the combination with more advanced local attention variants in future research.

`RAT` can be viewed as an attention mechanism with resolution, akin to how humans process information: short-term inputs are integrated into coherent representations, while long-term events are stored as memory anchors for selective retrieval. Compared to dilated attention [9], our simple recurrence enables global perception, and the chunk-level attention has high training parallelism and reduced cache memory (see Sec. C.1 for more). Finally, a recent work [46] also aggregates the key and value vectors and applies softmax-based attention over them. However, it compresses the full-sequence vectors into a 2D memory matrix, unlike our chunk-based design, which enables flexible memory slots.

## 6 Conclusion

This paper proposes the `RAT` structure, an intermediate architecture between RNN and attention. It segments sequences into chunks, applies RNN within each chunk to capture local dependencies, and employs attention across chunks to model long-range dependencies. We detail its architectural design and investigate its combination with the sliding-window attention mechanism. Experimental results across various settings demonstrate the efficiency and accuracy benefits of `RAT`, highlighting its potential for future language model development.

**Limitations.** Due to resource constraints and the exploratory nature of this work, our experiments are limited to models with up to 1.3B parameters. We have not yet scaled `RAT` to larger language models such as 7B or 14B to confirm our results hold at that scale. Additionally, we did not adopt supervised fine-tuning techniques commonly used in current industry practices, which typically involve substantial data resources and many engineering details. Instead, we followed a more classical approach based on train-test splits to evaluate performance, as our focus lies primarily on architectural design during pretraining. Lastly, `RAT` may still face length generalization issues from positional encoding, similar to attention. But as shown in Subsec. C.2, this can be significantly reduced by shorter inter-chunk attention spans, and we also explore the use of NoPE.

## Acknowledgments and Disclosure of Funding

We sincerely thank the RCP and the SCITAS team at EPFL for GPU support. We also thank the Swiss AI Initiative and the Swiss National Supercomputing Centre (CSCS) for supporting this work through grants under project IDs a06 and a10. We thank Skander Moalla, Anja Surina, and Lars Quaedvlieg for helpful discussion on SFT tasks. We extend our appreciation to Karin Getaz for administrative support.

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

# A  Implementation details

## A.1  Algorithm

We provide the pseudocode for the training and prefilling modes of RAT in Listing 1, and the pseudocode for the generation mode in Listing 2.

```python
def merge_last_token_stable
    (inter_out, intra_out, inter_lse, intra_lse):
    # Merge previous chunk's output and current chunk's output safely
    # by adjusting the base of the exponential in softmax
    max_lse = maximum(inter_lse, intra_lse)
    inter_lse_exp = exp(inter_lse - max_lse)
    intra_lse_exp = exp(intra_lse - max_lse)
    denom = intra_lse_exp + inter_lse_exp
    intra_adjust = (intra_lse_exp / denom).unsqueeze(-1)
    inter_adjust = (inter_lse_exp / denom).unsqueeze(-1)
    return inter_out * inter_adjust + intra_out * intra_adjust

def train_or_prefill
    (z, g, q, k, v, num_chunk, chunk_size, num_head, softmax_scale):
    """
    Args:
        z: (B, T, D), output gate after sigmoid function
        g: (B, T, D), forget gate after sigmoid function
        v: (B, T, D), value vectors in attention
        q, k: (B, T, D), shared query and key vectors in attention
    Returns:
        out: (B, T, D)
    """
    bs, seq_len, d_model = shape(z)
    k, v = [
    rearrange(m, "b (c l) d -> b c l d", c=num_chunk) for m in (k, v)]

    # Intra-chunk RNN: apply associative scan along the "l" dimension
    intra_v = ascan(g, (1.0 - g) * v)
    intra_k = ascan(g, (1.0 - g) * k)

    # Rearrange for multi-head attention
    intra_v
     = rearrange(intra_v, "b c l (a p) -> b a (c l) p", a=num_head)
    intra_k
     = rearrange(intra_k, "b c l (a p) -> b a (c l) p", a=num_head)
    q = rearrange(q, "b t (a p) -> b a t p", a=num_head)

    # Apply inter-chunk RoPE
    q, intra_k = apply_rope(q, intra_k)

    # Extract chunk's end representation for inter-chunk attention
    chunk_intra_k = intra_k[..., chunk_size - 1::chunk_size, :]
    chunk_intra_v = intra_v[..., chunk_size - 1::chunk_size, :]

    # Inter-chunk attention using flex_attention
    # with chunk-level causal mask and returning log-sum-exp scores
    block_mask = create_block_mask(
        lambda b, h, q_idx, kv_idx: q_idx // chunk_size > kv_idx,
        1, 1, q.shape[2], num_chunk
    )
    inter_out, inter_lse = flex_attention(
        q, chunk_intra_k, chunk_intra_v,
        scale=softmax_scale,
        block_mask=block_mask,
        return_lse=True
    )
```

```
54
55      # Compute logits with
         the current chunk's k, required separately due to causal masking
56      intra_lse
         = einsum("batp, batp -> bat", q, intra_k) * softmax_scale
57
58      # Merge outputs
         from previous chunks and current chunk (due to causal masking)
59      out =
         merge_last_token_stable(inter_out, intra_v, inter_lse, intra_lse)
60      out = rearrange(out, "b a t p -> b t (a p)", a=num_head)
61      return z * out
```

Listing 1: Pseudo code for the training or prefilling modes of RAT. We use Pytorch's *flex attention* and *associative scan* for implementation.

```
1   def gen(z, g, q, k, v, chunk_start, num_head, cache):
2       """
3       Args:
4           z: (B, 1, D), output gate after sigmoid
5           g: (B, 1, D), forget gate after sigmoid
6           v: (B, 1, D), value vector
7           q, k: (B, 1, D), shared query and key vectors across heads
8           chunk_start: int, index of current chunk
9           cache: tuple of four tensors
10              lastkcache
         : (B, 1, D), last token k in current chunk (for RNN-style update)
11              lastvcache
         : (B, 1, D), last token v in current chunk (for RNN-style update)
12              kcache: (B, A, C, P) or
          (B, num_head, num_chunk, head_dim): cached k at each chunk's end
13              vcache: (B, A, C, P) or
          (B, num_head, num_chunk, head_dim): cached v at each chunk's end
14          # Note: KVCache is stored per chunk, not per token.
15       Returns:
16           out: (B, 1, D)
17       """
18       lastkcache, lastvcache, kcache, vcache = cache
19
20       # Intra-chunk RNN: Recurrent update within the current chunk
21       intra_k = g * lastkcache + (1.0 - g) * k
22       intra_v = g * lastvcache + (1.0 - g) * v
23       lastkcache.copy_(intra_k)
24       lastvcache.copy_(intra_v)
25
26       # Rearrange input for attention
27       intra_k = rearrange(intra_k, "b t (a p) -> b a t p", a=num_head)
28       intra_v = rearrange(intra_v, "b t (a p) -> b a t p", a=num_head)
29       q = rearrange(q, "b t (a p) -> b a t p", a=num_head)
30
31       # Apply RoPE
32       q, intra_k = apply_rope(q, intra_k)
33
34       # Update the current chunk's kvcache
35       kcache[:, :, chunk_start:chunk_start + 1] = intra_k
36       vcache[:, :, chunk_start:chunk_start + 1] = intra_v
37
38       # Inter-chunk
         attention: standard attention over cached chunk representations
39       attn_out = scaled_dot_product_attention(
40           q, kcache[:, :,
         :chunk_start + 1], vcache[:, :, :chunk_start + 1], is_causal=False
41       )
42       out = rearrange(out, "b a t p -> b t (a p)", a=num_head)
```

```
43        return z * out
```

Listing 2: Pseudo code for the generation mode of `RAT`. We simply use the *flash attention* (Pytorch's one). Note that KVCache of `RAT` is reduced from $T$ to $C$ compared to the attention module.

## A.2 Experiments

**Dataset** In our preliminary studies, we use the PG19 dataset [47], a long-form English book corpus with inherently long contexts. For the 1.3B model experiments, we adopt the FineWeb-Edu dataset [26], using its 100B-token randomly sampled version downloaded from the HuggingFace repository. To match the pretraining context length, we concatenate documents using a separator token. Note that web samples are usually very short, compared to the book dataset. For downstream evaluation, we consider a suite of classical commonsense reasoning benchmarks from the Eleuther AI evaluation harness [28], including PIQA [48], ARC-C [49], and HellaSwag [50]. In the LLaMA2 tokenizer, we observe that the inputs of these tasks typically contain fewer than 300 tokens. For the LongBench evaluation, in Table 7 we provide input length for each task, offering a rough indication of task difficulty with respect to input length. For SFT-based tasks, we have elaborated sequence lengths in the main text. As LongBench and SFT tasks often involve very long inputs, we apply truncation in the middle to preserve information at both the beginning and the end.

Table 7: We report the average input length and the 95th percentile input length of each task, measured in tokens using the LLaMA2 tokenizer. NQA (NarrativeQA), MQA (MultiFieldQA-en), HQA (HotpotQA), WQA (2Wiki-MultihopQA), MSQ (Musique), GR (GovReport), MN (MultiNews), TQA (TriviaQA), and RBP (RepoBench-P).

| Task | Single-Document QA | | | Multi-Document QA | | | Summarization | | | Code Completion | |
|------|------|--------|------|------|------|------|------|-------|------|------|------|
| **Name** | NQA | Qasper | MQA | HQA | WQA | MSQ | GR | QMSum | MN | LCC | RBP |
| **Avg.** | 36037 | 5780 | 8115 | 15329 | 8483 | 18555 | 12280 | 15980 | 3156 | 4307 | 14818 |
| **95th pctl.** | 77966 | 10164 | 14994 | 19755 | 16939 | 20066 | 25721 | 29069 | 7135 | 10401 | 31937 |

**200M model** We start with a 200M-parameter model in our preliminary study, with a model dimension of 1024, 12 transformer layers, and head dimension of 64. The rotary position embedding (RoPE) base is set to 10,000. We use the GPT2 tokenizer. Following Mohtashami and Jaggi [45], we repeat the PG19 training split five times to reach a total of 15B training tokens. The learning rate is scheduled using cosine annealing, starting at $6.0 \times 10^{-4}$ and decaying to $1.0 \times 10^{-6}$, with a warm-up ratio of 10%. We use the AdamW optimizer with a weight decay of 0.1 and $\beta = (0.9, 0.98)$. Gradient clipping is applied with a threshold of 1.0, and the global batch size is set to 1M tokens. We explore training with three different context lengths: 8K, 16K, and 32K. Training these models on 4 H100 GPUs takes approximately 5 to 14 hours. In particular, the attention model requires up to 14 hours when the sequence length is $T = 32768$, whereas the `RAT(L=16)` model completes training in about 7 hours under the same setting.

**1.3B model** The 1.3B-parameter model uses a model dimension of 2048, 24 transformer layers, and a head dimension of 128, equipped with RMSNorm [51] and without bias. The RoPE base is also set to 10,000. The model parameters are initialized using a Gaussian distribution with a standard deviation of 0.02. We adopt the LLaMA2 tokenizer in the following studies. For pretraining, we use a cosine-annealed learning rate schedule starting at $8.0 \times 10^{-4}$ and decaying to $1.0 \times 10^{-6}$, with 5% warmup. The global batch size is set to 2M tokens, and the context window is set to 4096. Each model is trained on 16 H100 GPUs, requiring approximately 2 to 3 days to complete.

For LongBench evaluation, we follow the default prompts with greedy decoding for all tasks except summarization. For summarization, we apply a repetition penalty of 1.2 to address the common issue in pretrained-only models of generating repetitive outputs under instructional prompts. For SFT tasks, we train the models on the official training splits with an answer-only loss and evaluate them on the corresponding test sets. Although we explored different hyperparameters during the early experimentation, we observed that the relative trends remained largely stable. Thus, we fix the learning rate and batch size to $(1.0 \times 10^{-5}, 128)$ for large datasets, and $(1.0 \times 10^{-5}, 32)$ for the smaller QMSum [30] task, following common practice for 1B-scale models. The weight decay is

set as 0.01, and all other hyperparameters follow the pretraining setup. We sweep over the number of epochs and report the best result for each dataset and architecture. In practice, QA tasks typically converge in 1–2 epochs, while summarization benefits from slightly longer training.

# B  Efficiency

## B.1  Latency

To supplement Fig. 2 in main text, we put the concrete latency number of a single layer in Table 8, Table 9, and Table 10.

Table 8: Single temporal-mixing layer (including input and output projections) training time across different sequence lengths. The latency (ms) is tested on 200K tokens.

| Model | 4096 | 8192 | 16384 | 32768 | 65536 | 131072 | 262144 |
|---|---|---|---|---|---|---|---|
| RNN | 77.00 | 78.42 | 77.50 | 83.50 | 115.83 | 130.62 | 195.30 |
| RAT(L=128) | 150.19 | 150.76 | 156.65 | 154.52 | 159.94 | 165.07 | 206.46 |
| RAT(L=64) | 146.25 | 148.46 | 151.10 | 155.83 | 165.58 | 177.78 | 227.76 |
| RAT(L=16) | 150.08 | 153.63 | 158.98 | 173.89 | 202.00 | 249.79 | 378.11 |
| Attention | 122.06 | 176.44 | 277.61 | 474.90 | 877.14 | 1713.82 | 3417.48 |
| Attention/RAT(L=16) | 0.81× | 1.15× | 1.75× | 2.73× | 4.34× | 6.86× | 9.04× |

Table 9: Single temporal-mixing layer (including input and output projections) prefilling time across different sequences lengths. The latency (ms) is tested on 200K tokens.

| Model | 4096 | 8192 | 16384 | 32768 | 65536 | 131072 | 262144 |
|---|---|---|---|---|---|---|---|
| RNN | 24.93 | 24.75 | 25.32 | 27.09 | 30.54 | 39.40 | 56.16 |
| RAT(L=128) | 44.74 | 44.78 | 45.09 | 45.50 | 46.67 | 48.62 | 51.75 |
| RAT(L=64) | 43.46 | 44.27 | 44.27 | 44.83 | 47.08 | 50.83 | 57.38 |
| RAT(L=16) | 44.03 | 44.90 | 45.53 | 51.08 | 57.31 | 71.18 | 99.53 |
| Attention | 36.93 | 52.71 | 80.21 | 135.68 | 245.14 | 494.62 | 997.50 |
| Attention/RAT(L=16) | 0.84× | 1.17× | 1.76× | 2.66× | 4.28× | 6.95× | 10.02× |

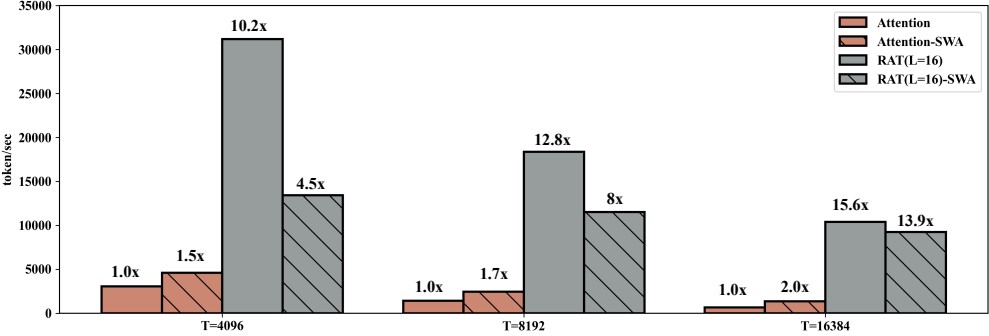

Figure 4: We measure the maximum throughput of the full 1.3B model for generating 1024 tokens under different prefilling lengths. For each total sequence length $T$, the prefilling length is set to $T - 1024$. For example, $T = 4096$ corresponds to a prefilling of 3072 tokens, while $T = 8192$ and $T = 16384$ correspond to 7168 and 15360 tokens, respectively. When the sequence length increases, the maximum throughput ratio between RAT and attention rises from $10.2×$ to $15.6×$, highlighting the strong efficiency advantage of RAT in long-context generation.

## B.2  Maximum throughput

As RAT also reduces cache memory usage, we report the maximum throughput of the full 1.3B model across different sequence lengths in Fig. 4. For example, RAT(L=16) achieves an approximately

Table 10: Single temporal-mixing layer (including input and output projections) generation time at the specified position. The latency (ms) is tested on generating batches of tokens with $B=64$, $B=512$, and $B=1024$.

| Model | 4096 | 8192 | 16384 | 32768 | 65536 | 131072 | 262144 |
|---|---|---|---|---|---|---|---|
| $B=64$ | | | | | | | |
| RNN | 0.36 | 0.33 | 0.34 | 0.33 | 0.33 | 0.33 | 0.33 |
| RAT(L=128) | 0.62 | 0.60 | 0.61 | 0.63 | 0.66 | 0.66 | 0.68 |
| RAT(L=64) | 0.67 | 0.64 | 0.66 | 0.70 | 0.73 | 0.74 | 1.03 |
| RAT(L=16) | 0.62 | 0.65 | 0.64 | 0.68 | 1.04 | 1.82 | 3.42 |
| Attention | 1.15 | 1.73 | 3.33 | 6.66 | 12.94 | 26.8 | OOM |
| $B=512$ | | | | | | | |
| RNN | 0.35 | 0.34 | 0.33 | 0.33 | 0.34 | 0.34 | 0.33 |
| RAT(L=128) | 0.63 | 0.64 | 0.66 | 0.75 | 1.17 | 2.44 | 3.46 |
| RAT(L=64) | 0.70 | 0.68 | 0.74 | 1.16 | 2.41 | 3.44 | 6.69 |
| RAT(L=16) | 0.75 | 1.22 | 2.45 | 3.48 | 6.73 | 13.09 | 25.76 |
| Attention | 6.56 | 12.89 | 25.84 | OOM | OOM | OOM | OOM |
| $B=1024$ | | | | | | | |
| RNN | 0.36 | 0.36 | 0.35 | 0.35 | 0.35 | 0.36 | 0.35 |
| RAT(L=128) | 0.73 | 0.75 | 0.95 | 1.35 | 2.54 | 4.64 | 6.98 |
| RAT(L=64) | 0.74 | 0.94 | 1.35 | 2.48 | 4.67 | 7.02 | 13.45 |
| RAT(L=16) | 1.38 | 2.56 | 4.63 | 7.04 | 13.51 | 26.43 | OOM |
| Attention | 13.04 | 25.70 | OOM | OOM | OOM | OOM | OOM |

$10.2\times$ higher throughput than the baseline attention model at $T=4096$, and improves further to $15.6\times$ at $T=16384$. Similarly, RAT(L=16)-SWA reaches $4.5\times$, $8.0\times$, and $13.9\times$ higher throughput over attention at $T=4096$, 8192, and 16384, respectively.

## C  Accuracy

### C.1  Supplementary ablation study

All experiments here are conducted on the PG19 Book dataset with 200M-parameter models, and FLOPs are controlled for comparison.

**Reversed design of RAT**  Instead of employing recurrence within each chunk, we also explored a reversed design that applies attention inside the chunk, followed by recurrence to capture long-range dependencies. For this reversed design, standard RoPE is sufficient. Regarding the parameter allocation, we found it crucial to retain the key vector in the attention module as a full tensor, as sharing it together with the query vector leads to a collapse into single-head attention. Thus, the final reversed design shares the query vector across attention heads, employs low-rank matrices for the projection used for output gate, and preserves full projections for both the forget gate and the key vector.

Results of the reversed design are put in Table 11. It can be observed that, under identical FLOP constraints, this design significantly underperforms RAT. We think this is because short-context dependencies are easier to capture and do not suffer from memory degradation, so using lightweight recurrence locally is more efficient. Long-context dependencies are harder and may require direct retrieval of distant information, where attention is more suitable. For example, to match the FLOPs budget of $\mathcal{O}(T/64)$, the reversed design requires chunk sizes of 128, 256, and 512 for sequence lengths of 8192, 16384, and 32768, respectively. With this small local window size, the long-range dependencies are attributed to the recurrence, thus leading to high perplexity. Only when we reduce the number of chunks to 16 does the perplexity drop below 14.00. Therefore, while the reversed design also allows for interpolation between attention and RNN (reducing to attention when $L=T$, and to RNN when $L=1$), we choose to focus on RAT due to its more efficient utilization of FLOPs.

**Comparison to FLASH [10]** As pointed out in Sec. 5, Hua et al. [10] also uses the concept of chunking; however, we adopt a fundamentally different framework, both in the components involved and in how the two outputs are combined. Specifically, Hua et al. [10] simply adds the outputs of inter-chunk and intra-chunk computations, whereas RAT organizes them hierarchically, applying recurrence over the key and value before inter-chunk attention. We argue that these differences lead to the weaker performance of Hua et al. [10], as shown in Table 13. First, as discussed above, RAT utilizes FLOPs more effectively than methods based on local attention, since its long-range dependencies must rely on memory-degrading components such as recurrence and linear attention. Second, we find that directly adding the outputs of softmax-based attention and linear attention introduces differences in output scale and potential representational conflicts, which may lower down the performance.

**Comparison to dilated attention** We discuss the difference between RAT and the design that interleaves recurrence and dilated attention [9] with the sliding pattern. First, from an efficiency perspective, RAT has advantages over dilated attention in both training and inference. During training, dilated attention reduces parallelism by the dilation rate, since tokens within the same dilation span attend to different KV vectors. In contrast, RAT allows tokens to share the same chunk-level representation, except within their own chunk. During inference, the sliding dilation pattern causes a memory issue, since the KV cache of all tokens must be stored, whereas RAT reduces the cache size proportionally to the chunk size. Second, in terms of accuracy, to match the FLOPs, we set the dilation rate to 64 for RNN-Dilated attention, since dilated attention occurs in only half of the layers. It can be seen from Table 14 that dilated attention performs poorly. We observed significantly slower convergence at the beginning of training, likely due to its lack of global perception.

Table 11: Perplexity results of the reversed design, where attention is applied within chunks and RNN is used across chunks. Experiments are conducted on the 200M model trained on the PG19 book dataset. Under the same FLOPs, it can be observed that RAT significantly outperforms its reversed counterpart.

Table 12: Pretraining and downstream evaluation results of 1.3B models using either RoPE or NoPE positional encodings. RoPE is used in the main text, while NoPE is trained for investigating length extrapolation. Notably, NoPE achieves reasonable performance even at the 1B model scale with only 100B training tokens.

| Method | FLOPs | T=8192 | T=16384 | T=32768 |
|---|---|---|---|---|
| RAT(L=128) | $\mathcal{O}(T/128)$ | 13.57 | 13.44 | 13.66 |
| RAT(L=64) | $\mathcal{O}(T/64)$ | 13.42 | 13.34 | 13.39 |
| RAT-Reversed(C=128) | $\mathcal{O}(T/128)$ | 14.53 | 14.35 | 14.37 |
| RAT-Reversed(C=64) | $\mathcal{O}(T/64)$ | 14.21 | 14.06 | 14.02 |
| RAT-Reversed(C=16) | $\mathcal{O}(T/16)$ | 13.69 | 13.48 | 13.50 |
| Attention | $\mathcal{O}(T)$ | 13.26 | 13.10 | 13.19 |

| Method | PE | Pretrain PPL | HellaSwag acc_norm | LAMBADA acc | PIQA acc_norm |
|---|---|---|---|---|---|
| Attention-SWA | RoPE | 7.61 | 57.1 | 48.0 | 72.7 |
| RAT(L=16)-SWA | RoPE | 7.57 | 57.9 | 49.3 | 73.9 |
| Attention-SWA | NoPE | 7.69 | 56.4 | 47.3 | 72.7 |
| RAT(L=16)-SWA | NoPE | 7.63 | 56.8 | 47.8 | 73.7 |

Table 13: Performance of different chunk-based designs with $T = 16384$ and FLOPs $\mathcal{O}(T/64)$.

Table 14: Performance compared to dilated attention with sliding patterns under the same FLOPs.

| Method | Intra-chunk | Inter-chunk | Organization | PPL |
|---|---|---|---|---|
| RAT(L=64) | Recurrence | Attention | Hierarchically | **13.34** |
| RAT-Reversed(C=64) | Attention | Recurrence | Hierarchically | 14.06 |
| FLASH (C=128) | Attention | Linear attention | Add together | 14.4 |

| Method | PPL |
|---|---|
| RNN-Dilated attention | 15.22 |
| RAT(L=128) | 13.44 |

## C.2 Length generalization

Because of the use of softmax-based attention and RoPE at the inter-chunk level, it is reasonable to expect that RAT may also face challenges in length generalization, where the model is trained on short sequences but evaluated on much longer ones. To study this, we consider SWA variants using either RoPE or NoPE for attention and RAT, following recent practices that interleave attention with local attention modules and apply positional encodings only in the local attention. This design has been adopted in practice and has shown promising results [22].

As shown in Fig. 5, the RNN performs very steadily as the test sequence length increases. However, within the training context, it has the highest loss among all models. With RoPE, both Attention-SWA and RAT(L=16)-SWA experience a sharp increase in loss when the sequence length goes beyond 6000. The increase is more severe in the attention model than in the RAT variant. This is likely because RAT reduces the effective attention span by attending only to inter-chunk positions, which may offer better robustness to extrapolation.

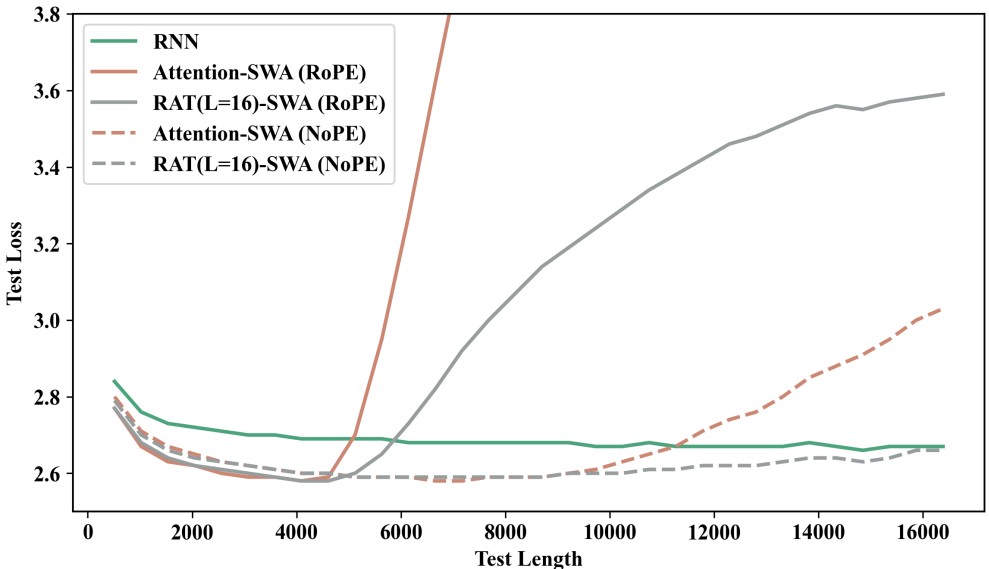

Figure 5: Evaluation at different test lengths for pretrained models trained with a 4K context window. `RAT(L=16)-SWA` with NoPE achieves the best overall performance, exhibiting strong generalization up to $T = 16384$ while maintaining low loss within the training context.

We also pretrain NoPE versions, where no positional encodings are used in attention or `RAT` layers. Interestingly, these NoPE models show reasonable performance, as reported in Table 12. While their pretraining perplexities are higher than those of the RoPE variants, the gap remains small, especially considering the 1.3B model size and 100B-token training budget. To further verify this, we evaluate the models on several commonsense reasoning tasks and obtain promising results. We expect that NoPE will perform even better with larger models and longer training. In terms of length extrapolation, using NoPE significantly improves the robustness for both Attention-SWA and `RAT(L=16)-SWA`. Among them, `RAT(L=16)-SWA` with NoPE shows the most stable performance, maintaining low loss even at a sequence length of 16,384.

In conclusion, we find that `RAT` can outperform the attention module in length generalization, both with RoPE and with NoPE. Still, the problem is not fully solved, as none of the models reach the stability level of the simple RNN. While many techniques have been proposed to improve length extrapolation in attention models, they should also be considered for `RAT`, as `RAT` essentially redirects attention from every position to inter-chunk locations. For example, RoPE extrapolation in the attention module can be improved using methods from Peng et al. [52], bloc97 [53], and NoPE extrapolation has been studied in Wang et al. [54], which points out that NoPE still has a context length limit, though it performs better than RoPE. Their work links the failure to shifts in attention distributions and proposes tuning the temperature of attention heads to improve extrapolation. We believe such methods could also be applied to `RAT`, and leave further exploration of these techniques to future work.

### C.3 Needle-in-Haystack (retrieval ability)

To evaluate retrieval capabilities—an area where attention-based architectures are known to excel, and where RNNs typically fall short—we conduct experiments on the Needle-in-Haystack tasks. In these tasks, a "needle" (e.g., a magic number or UUID) is embedded within irrelevant passages, noisy text, or mixed with multiple key-value pairs. Each key-value pair is presented as a short identifier (the key) followed by its associated value, and the model is prompted to retrieve the correct value given a specific key.

We adopt the RULER benchmark [32] and evaluate a range of Needle-in-Haystack task configurations. Specifically, task `single_1` involves retrieving a specific number from a background filled with repeated noise. In `single_2`, the background is replaced with natural stories. `single_3` increases the difficulty by requiring retrieval of a long and complex UUID. The `multikey` tasks are more

challenging than the single-key ones, as they introduce multiple such key-value pairs into the context. In particular, `multikey_2` and `multikey_3` consist almost entirely of densely packed key-value pairs, among which only a single key is queried at the end. This makes the task especially challenging, as the model must retrieve the correct item from a highly cluttered input full of distractors. And `multikey_3` further incorporates the complex UUID format. In the `multiquery` and `multivalue` settings, the model is required to resolve multiple retrieval targets, either by answering several distinct queries or by retrieving all values associated with a single key. For detailed definitions of each task, we refer the reader to the benchmark documentation.

During evaluation, we observed that models may fail to interpret certain prompts correctly. For instance, prompts like *"A special magic number is hidden within the following text. Make sure to memorize it. I will quiz you about the number afterwards."* can cause failures even in attention-based models, especially on the harder `multikey` tasks. To mitigate this, we apply a light, one-round supervised fine-tuning stage before evaluation to adapt the models to the instruction patterns. We generate 1000 synthetic training samples for each of the 8 tasks, resulting in a total of 8000 examples disjoint from the validation sets. The models are trained on this dataset for one round and then are evaluated directly on all 8 tasks. This procedure yields a fairer and more stable comparison. As shown in Table 6, RNNs perform reasonably well on the simpler tasks (`single_1`, `single_2`), but their performance drops to near zero on the harder ones. Attention-based models perform consistently well, benefiting from their ability to access all previous tokens directly. `RAT(L=16)` achieves results close to full attention, especially in numeric tasks. However, it struggles on UUID tasks due to their complexity and the use of exact match scoring (`single_3`, `multikey_3`). Meanwhile, `RAT(L=64)` falls between RNNs and `RAT(L=16)`, as expected given its partial access to long-range context.

These results are consistent with the underlying architecture designs: attention provides full direct access to all past tokens, RNNs compress all past information into a hidden state, while `RAT` compresses part of the history but also retains direct access at the chunk level. As a result, `RAT` naturally exhibits retrieval capabilities that lie between those of attention and RNNs.

## D  Broader Impacts

Enhancing the efficiency of Large Language Models (LLMs) can significantly reduce computational resources and energy consumption, benefiting the environment and democratizing access to advanced AI technologies. However, increased efficiency could also lead to greater dissemination of disinformation and the creation of deepfakes, posing risks to public trust and security and potentially reinforcing existing biases that impact specific groups unfairly. This research aims to promote the responsible development and deployment of LLMs, maximizing societal benefits while acknowledging potential harms.

## E  License information

- FineWeb-Edu (dataset): Open Data Commons License Attribution family.
  Link: `https://huggingface.co/datasets/HuggingFaceFW/fineweb-edu`
- LongBench (dataset and code): MIT License.
  Link: `https://github.com/THUDM/LongBench`
- NarrativeQA (dataset): Apache License 2.0.
  Link: `https://github.com/deepmind/narrativeqa`
- QMSum (dataset): MIT License.
  Link: `https://github.com/Yale-LILY/QMSum`
- WikiSum (dataset): Custom license (unspecified).
  Link: `https://huggingface.co/datasets/d0rj/wikisum`
- lm-evaluation-harness (code): MIT License.
  Link: `https://github.com/EleutherAI/lm-evaluation-harness`
- RULER benchmark (code): Apache 2.0 License.
  Link: `https://github.com/NVIDIA/RULER`

