# OpenReview forum: "RAT: Bridging RNN Efficiency and Attention Accuracy via Chunk-based Sequence Modeling"
_NeurIPS.cc/2025/Conference — NeurIPS 2025 poster_

### Official Review · Reviewer_FAfY · 2025-06-30

**Clarity:** 3
**Significance:** 3
**Originality:** 3
**Rating:** 4
**Confidence:** 4

**Summary:**

The paper introduces RAT (Recurrent-Attention Tradeoff), a novel architecture designed to bridge the efficiency of RNNs with the accuracy of attention mechanisms in language modeling. RAT segments input sequences into chunks, applying a lightweight linear recurrence within each chunk for local modeling and using softmax attention across chunks to capture long-range dependencies. This hybrid structure enables flexible trade-offs via chunk size, interpolating between RNN-like and attention-like behavior. The authors demonstrate that RAT significantly improves training and inference speed, up to 10×, without substantial loss in accuracy. Through extensive pretraining and evaluation at the 1.3B parameter scale, RAT matches or exceeds standard attention on various tasks, especially in long-context settings. A further enhancement interleaves RAT with sliding-window attention to boost local precision, showing consistent performance gains across commonsense reasoning, code, and summarization tasks. The study highlights RAT's efficiency, architectural simplicity, and strong empirical results, positioning it as a promising alternative for scalable language models.

**Questions:**

1. How well does RAT scale to larger models like 7B or 13B? Please comment on any challenges with parallelism, memory, or throughput at scale. Concrete results or estimates would help assess broader relevance.

2. RAT underperforms on tasks like WinoGrande and few-shot QA. Can you explain whether this is due to chunking, attention granularity, or training setup? A deeper analysis would clarify limits of generalization.

3. The reversed design (attention within chunk, recurrence across) is said to perform worse. Can you provide more insight or experiments to show why? Understanding this failure would help validate your design choices.

4. How does RAT compare directly to recent efficient models like Mamba or DeltaNet under similar compute or FLOP budgets? A head-to-head baseline is needed to contextualize the claimed efficiency and accuracy trade-offs.

**Ethical Concerns:**

["NO or VERY MINOR ethics concerns only"]

**Final Justification:**

I am satisfied with the author's response and have updated my evaluation slightly upward to reflect the clarifications provided.

**Limitations:**

yes

**Paper Formatting Concerns:**

-

**Quality:**

3

**Strengths And Weaknesses:**

Quality: The paper presents a technically solid and well-executed study, with claims supported by strong empirical results across pretraining, evaluation, and fine-tuning settings. The proposed RAT architecture is carefully implemented and benchmarked against attention and RNN baselines. However, theoretical analysis is limited, and some performance gaps (e.g., in tasks like WinoGrande or few-shot QA) are acknowledged but not deeply investigated. The work is complete and honestly evaluated, though larger model scaling and ablations at higher capacity would strengthen the evidence.

Clarity: The paper is mostly clear and well-organized, with helpful diagrams and consistent terminology. Key ideas such as intra-chunk recurrence and inter-chunk attention are explained effectively. However, some descriptions (e.g., implementation efficiency or positional encoding strategies) are overly condensed and require more unpacking. Presentation could be improved with clearer separation between core contributions and peripheral optimizations.

Significance: RAT targets a critical challenge in scaling language models efficiently, and the results suggest meaningful practical benefits in speed and memory while maintaining accuracy. The ability to interpolate between attention and recurrence is valuable, and the hybrid design with sliding-window attention shows promise. While the impact at larger scales remains untested, the demonstrated improvements at 1.3B suggest the approach is relevant and could influence future model design.

Originality: The idea of chunked recurrence combined with sparse cross-chunk attention is a novel architectural middle ground, distinct from both state-space models and purely linear attention. While the components are not new individually, their integration is original and motivated by practical trade-offs. The paper provides a new way to think about temporal mixing that departs from both strict attention and RNN paradigms, though more conceptual framing and discussion of prior hybrid efforts would make the novelty clearer.

---

> ### Author Rebuttal · Authors · 2025-07-31
>
> We would like to thank the reviewer for the constructive feedback. Here are our responses.
> > Q1: How well does RAT scale to larger models like 7B or 13B? Please comment on any challenges with parallelism, memory, or throughput at scale. Concrete results or estimates help assess broader relevance.
>
> A1: This is an important question. We did not train RAT on 7B or 13B models because we prioritized model and architecture designs in this paper, especially given the compute resources available to us. However, we provide efficiency analyses and results at larger model scales to demonstrate its potential.
>   * For parallelism, we mentioned in the paper that RAT is compatible with both tensor parallelism and context parallelism. Both the recurrence and cross-chunk attention are head-independent, enabling standard tensor parallelism by assigning different heads to different GPUs. For context parallelism, the within-chunk recurrence is chunk-independent and works well with small chunk sizes, allowing chunks to be distributed across GPUs. Since RAT stores only a small number of key/value vectors (e.g., 16× fewer), cross-chunk attention may even avoid ring-style communication.
>   * For compute-intensive training, we benchmarked the temporal mixing block (including input/output projections) and found that speed-up decreases with larger model dimension D, as linear projections dominate FLOPs more. For instance, when D increases from 2048 to 4096, the speed-up drops from 7× to 5.5× (Attention block: 1776.51s vs. RAT(L=16) block: 322.43s for 100K tokens). However, training a 7B model typically involves tensor parallelism, which reduces the per-GPU dimension (e.g., D = 2048 with 2 GPUs), allowing the 7× speed-up to be retained.
>   * For memory-intensive decoding, scaled-dot product attention remains a major bottleneck, even in large models. We benchmark latency on a single temporal mixing block when generating tokens at a specific position, and measure the maximum throughput of full models when generating 1024 tokens with a 3072-token prompt. RAT consistently outperforms attention, and on the 13B model, where attention suffers from limited batch size and poor GPU utilization, RAT achieves even higher improvements.
>
> |Latency of Temporal mixing block (D=4096)|Position=4096, Batch=1024|P=8192, B=512|P=16384, B=256|P=32768, B=128|P=65536, B=64|
> |-|-|-|-|-|-|
> | RAT(L=16)| 3.01s| 2.7s|2.25s| 2.21s| 2.15s  |
> | Attention|26.16s| 25.82s|25.87s| 25.8s| 26.22s |
> | Ratio|8.7x|9.6x|11.5x|11.7x| 12x |
>
> |Maximum throughput of a whole model (token/sec)|1.3B (results in our paper)|7B|13B|
> |-|-|-|-|
> |RAT(L=16)|31170|10103|5749|
> |Attention|3152|983|534|
> | Ratio|10.2x|10.3x|10.8x |
>
> > Q2: How does RAT compare directly to recent efficient models like Mamba or DeltaNet under similar compute or FLOP budgets? A head-to-head baseline is needed to contextualize the claimed efficiency and accuracy trade-offs.
>
> A2: Thanks for the suggestion. Since RAT is a novel idea with a simple structure that interpolates between RNN and attention, we primarily compared it with strong attention and RNN baselines for clarity and showed comparable performance to attention. We also noted in lines 245–246 and the footnotes that our best model outperforms the best model in a recent linear attention paper [1]. Below we provide detailed comparisons with these models and will include them in the revision.
>
> [1] trained these models with the same model size (1.3B), sequence length (4096), and datasets (fineweb_edu 100B) as ours. We calculate the value/state size based on task sequence length T, dimension D, and number of layers N. For state space models, we multiply the value dimension by the state expansion factor and note that Mamba2 and GatedDeltaNet expand the value dimension. Since these commonsense reasoning tasks contain very short contexts (≤300 tokens), RAT(L=16) only has at most 20ND value vectors, yet it surpasses others in 4 of 7 tasks, and RAT-SWA outperforms GatedDeltaNet-SWA in 6 of 7 tasks. On retrieval-heavy tasks (Needle-in-Haystack and LongBench QA), RAT also consistently outperforms them.
>
> [1]. Gated Delta Networks: Improving Mamba2 with Delta Rule. ICLR’25
> |Model|value/state size (T=300, D=2048, N=24) |arc_challenge|arc_easy|hellaswag|lambada|piqa|winogrande|boolq|
> |-|-|-|-|-|-|-|-|-|
> |||acc/acc_n|acc/acc_n|acc/acc_n|acc|acc/acc_n|acc|acc|
> |Mamba2|256ND|-/37.88| 72.47/-|-/55.67|45.66|71.87/-|55.24|60.13|
> |DeltaNet|128ND|-/35.66| 68.47/-|-/50.93|42.46|70.72/-|53.35|55.29|
> |GatedDeltaNet| 288ND| -/38.39|71.21/- |-/55.76|**46.65**|**72.25**/-|**57.45**|60.24|
> |RAT(L=16)|~20ND| 36.35/**39.33**|**72.64**/67.80| 43.27/**56.44**|44.4|72.03/72.69|53.2|**63.3**|
> |Hybrid|
> |GatedDeltaNet-SWA(2048)|288N/2D-300N/2D|-/40.10|71.75/-|-/56.53|47.73|72.57/-|**58.4**|63.21|
> |RAT(L=16)-SWA(1024)|~20N/2D-300N/2D| 37.20/**40.78**|**72.56**/68.22| 44.33/**57.85**|**49.33**|**73.29**/73.94| 56.91|**63.21**|
>
> |Model|value/state size (T=4096, D=2048, N=24)|NIAH-1|NIAH-2|NIAH-3|
> |-|-|-|-|-|
> |DeltaNet|128ND|**99**|18.6|22.4|
> |Mamba2|256ND|65.4|56.2|4.6|
> |Gated DeltaNet|288ND|91.4|92.2|27.6|
> |RAT(L=16)|256ND|97|**97.6**|**56.4**|
>
> |Model|value/state size (T=4096, D=2048, N=24) |NQA|Qasper|MQA|HQA|WQA|MSQ|
> |-|-|-|-|-|-|-|-|
> |Mamba2|256ND|11.1|11.3|18.6|11.8|15.1|**6.7**|
> |DeltaNet|128ND|12.9|10.8|21.5|10.9|13.2|5.1|
> |GatedDeltaNet|288ND|14.1|14|23.3|13.7|14.1|5.8|
> |RAT(L=16)|256ND|**14.5**|**16.1**|**28.3**|**16.7**|**18.9**|6.3|
>
> > Q3: The reversed design (attention within chunk, recurrence across) is said to perform worse. Can you provide more insight or experiments to show why?
>
> A3: We provided relevant discussion and results in Appendix C.1 and Table 9. We found that the reversed design uses FLOPs less effectively. Short-context dependencies are easier to capture and do not suffer from memory degradation, so using lightweight recurrence locally is more efficient.  Long‑context dependencies are harder and may require direct retrieval of distant information, where attention is more suitable.
>
> In the reversed design, maintaining good performance requires a large chunk size; otherwise, the recurrence must handle long-term history alone and may suffer from memory degradation. But with large chunk sizes, most FLOPs are spent computing local information. For example, Table 9 shows that on the Book dataset (sequence length 8192), the reversed design needs a chunk size of 512 (with 16× FLOPs reduction) to reach low perplexity. In contrast, RAT achieves even lower perplexity with a chunk size of 64 and 64× FLOPs reduction (128 chunks for inter-chunk attention).
>
> Overall, RAT uses FLOPs more effectively. That said, the two designs may be complementary with RAT being better at preserving long-term information, and the other emphasizing local modeling. Due to resource constraints, we only explored interleaving RAT with local attention (without recurrence across chunks) in our large-scale experiments and leave combining both designs for future work.
> > Q4: RAT underperforms on tasks like WinoGrande and few-shot QA. Can you explain whether this is due to chunking, attention granularity, or training setup?
>
> A4: We think the weak performance of RAT (L=16) on WinoGrande is due to task and model instability in small‑scale pre‑training. The evaluation set contains only ~1k items and it is a binary choice task, so biases in the small-scale pretraining on this task can easily push accuracy to 50–60.  Another task we did not include but found a similar problem in the paper is BoolQ, where attention performs poorly (62.54 for RNN, 63.30 for RAT, and 58.23 for attention).
>
> As for the weak performance of RAT (L=16)-SWA on TREC (results on other few-shot QA tasks are comparable, so we focus on TREC here), we think the main reason is that LongBench frames this classification task with 50 mixed coarse‑ and fine‑grained labels in a few‑shot task. Without post‑training or instruction tuning, models may generate semantically correct but non‑exact answers and be uncertain about the required specificity. RAT (L=16)-SWA often produces more descriptive outputs for some labels (e.g., “Date of holiday” for “Date”), and simply correcting these would bring its performance close to the attention and RAT. For location‑related questions, which form a large portion of the test set, all models except attention‑SWA tend to predict the generic “Other location” instead of fine‑grained labels like “City,” explaining attention‑SWA’s higher accuracy on TREC.
> > Q5: The paper provides a new way to think about temporal mixing that departs from both strict attention and RNN paradigms, though more conceptual framing and discussion of prior hybrid efforts would make the novelty clearer.
>
> A5:  We’d like to first clarify that RAT is not a hybrid model. It applies the same mechanism to all tokens, heads, and layers. But hybrid models use different modules like attention or state-space in different places. Hybrid strategies are orthogonal to RAT. In our paper, we explored interleaving RAT with local attention and leave other directions, such as varying chunk sizes across layers or heads, for future work.
>
> Regarding novelty, one key contribution is the chunk-based design. It allows partial compression of history while preserving direct access to past tokens, which helps retrieve distant information, especially in noisy contexts. Unlike recent state space models that has fixed memory slots and may suffer from memory degradation on long inputs, RAT’s memory slots grows with sequence length while maintaining a fixed FLOPs reduction ratio. Finally, while local attention has been widely adopted in industry, RAT can complements it well by effectively and efficiently preserving long-term information.
>
> We will make these points clearer in the revision.
> >Q6: Presentation could be improved with clearer separation between core contributions and peripheral optimizations.
>
> A6: Thanks for pointing this out. We'll update the revision based on the suggestion.

---

> > ### Author Response · Authors · 2025-08-06
> > **Discussion phase closing soon**
> >
> > Dear Reviewer FAfY,
> >
> > Thank you again for your constructive feedback.  As we are getting closer to the end of the discussion period, could you please let us know if our responses above have adequately addressed your concerns?
> >
> > If you have any further questions or concerns, we would be happy to clarify. We sincerely appreciate the time and effort you have devoted to reviewing our work, and we look forward to your response.

---

### Official Review · Reviewer_pHG2 · 2025-07-01

**Clarity:** 3
**Significance:** 1
**Originality:** 2
**Rating:** 2
**Confidence:** 4

**Summary:**

Attempts to bridge the efficiency of RNNs (actually gated linear recurrence) and the accuracy of attention mechanisms for language modeling.
 - RAT divides input sequences into chunks, applying lightweight gated linear recurrence within each chunk to capture local dependencies and softmax attention across chunks for long-range interactions.
- RAT interpolates between RNN and attention behaviors, balancing computational cost and performance.
- RAT achieves significant efficiency gains: up to 7× training speedup for 1M-token sequences and 9× generation speedup at 4K tokens, with reduced KV cache usage
- Empirically, RAT matches or surpasses standard attention on benchmarks like LongBench (e.g., 24.9 vs. 18.6 on GovReport) and commonsense reasoning tasks.
- A hybrid RAT-SWA variant, interleaving RAT with sliding-window attention, further boosts performance (e.g., +4 points on code tasks, +1 Rouge-L on summarization).

**Questions:**

- I am surprised at the 7x training speedup and 9x generation speedup, and I am curious what the baseline comparisons are. Was flash attention used?

**Ethical Concerns:**

["NO or VERY MINOR ethics concerns only"]

**Final Justification:**

I've made minor modifications to the rating based on the author response, but ultimately there are some fundamental limitations of the paper.

**Limitations:**

None.

**Paper Formatting Concerns:**

None.

**Quality:**

2

**Strengths And Weaknesses:**

Strengths:
- Achieves up to 7× training speedup (1M tokens) and 9× generation speedup (4K tokens), with reduced KV cache usage, minimizing OOM risks. However, these efficiency gains may fade at a larger scale when MLP computation dominates.

Weaknesses:
- One big issue I have is that Eqn. 2. is a specific case of Mamba, and that this proposal for a new sequence mixing method is a mix of attention and this EMA construction.
  - A more meaningful comparison then should also be between Mamba-Transformer Hybrid architectures, both in terms of performance on downstream tasks, and on efficiency metrics.
- Because of the EMA-style construction of the "RNN" part, this also does not afford the model the capability to do state-tracking: https://arxiv.org/pdf/2404.08819.
   Since I suspect the speedups gained in the paper may fade with 1) different implementation comparisons, or 2) scale of model, I think there is little gain in capability if we consider the gains from using a RAT-Attention hybrid vs a Mamba-Attention hybrid.

---

> ### Author Rebuttal · Authors · 2025-07-31
>
> We thank the reviewer for the feedback. Here are the responses.
>
> > Q1:  One big issue I have is that Eqn. 2. is a specific case of Mamba, and that this proposal for a new sequence mixing method is a mix of attention and this EMA construction. A more meaningful comparison then should also be between Mamba-Transformer Hybrid architectures, both in terms of performance on downstream tasks, and on efficiency metrics.
>
> A1: We would like to clarify an important point which might have been misunderstood for RAT. RAT first chunks the inputs, applies recurrence within each chunk to aggregate “key” and “value” vectors, and then applies cross-chunk nonlinear attention to query chunk-level “key” and “value”. This same mechanism is used across all layers, heads, and tokens, making RAT not a hybrid model, whereas Mamba-Transformer alternates state-space modules and attention in different layers. And we're not putting the EMA construction in one layer and placing the attention in the second layer. In detail,
>
>   * The way recurrence used in RAT is different from Mamba and can be extended to different kinds of RNN in the future. Eqn. (2) in RAT is applied only within chunks and over both “key” and “value”. This chunking design allows the simple recurrence to work well on short chunks, and enables future exploration of more advanced RNNs, such as 2D recurrence for large chunks or nonlinear RNNs for small chunks. By contrast, Mamba-1 aggregates the “value” over the entire sequence (which can suffer from memory degradation issues on very long inputs) and is limited to linear RNNs. Mamba-2 cannot be reduced to Eqn. (2), as it is the linear attention with per-head gating.
>   * RAT’s cross-chunk attention is applied sparsely over previous chunks, reducing FLOPs compared to standard attention. We also designed an efficient implementation using FlexAttention to handle causal masking during training.
>
> Since RAT interpolates between attention and RNN layers, we mainly compared against these baselines in the paper and noted (lines 245–246, footnotes) that RAT outperforms recent linear attention works. Any hybrid strategy (e.g., mixing mechanisms in different layers or heads) are indeed orthogonal to RAT. In this paper, we explored RAT-SWA, which interleaves RAT with sliding-window attention. We now post comparisons with other state-space models below.
> | Hybrid models | value/state size (T=300, D=2048, N=24 (number of layers)) | arc_challenge | arc_easy     | hellaswag    | lambada | piqa  | winogrande | boolq |
> |-|-|-|-|-|-|-|-|-|
> |||acc/acc_n|acc/acc_n|acc/acc_n|acc|acc/acc_n|acc|acc|
> |Mamba1|64ND|-/35.40|69.52/-|-/52.91|43.98|71.32/-|52.95|61.13|
> |Mamba2|256ND|-/37.88| 72.47/-|-/55.67|45.66|71.87/-|55.24|60.13|
> |DeltaNet|128ND|-/35.66| 68.47/-|-/50.93|42.46|70.72/-|53.35|55.29|
> |GatedDeltaNet| 288ND| -/38.39|71.21/- |-/55.76|**46.65**|**72.25**/-|**57.45**|60.24|
> |RAT(L=16)|~20ND| 36.35/**39.33**|**72.64**/67.80| 43.27/**56.44**|44.4|72.03/72.69|53.2|**63.3**|
> |Hybrid|
> |Samba (Mamba1-SWA(2048) | 64N/2D-300N/2D| -/36.17| 68.81/- | -/53.42 | 44.94   | 70.94/- | 55.56 |62.11|
> |GatedDeltaNet-SWA(2048) | 288N/2D-300N/2D| -/40.10 | 71.75/-   | -/56.53 | 47.73   | 72.57/- | **58.4** |63.21|
> |RAT(L=16)-SWA(1024)| ~20N/2D-300N/2D| 37.20/**40.78**| **72.56**/68.22  | 44.33/**57.85** | **49.33**|**73.29**/73.94 | 56.91| **63.21**|
>
> Results for other models are taken from a recent paper [1], where they were trained with the same model size, datasets, and sequence lengths as ours. These commonsense reasoning tasks have short inputs (≤300 tokens), so RAT with a chunk size of 16 has at most \~20 chunks (\~20ND value vectors), yet it achieves better performance with much smaller “value” size.  The RAT-SWA also surpasses other Mamba-SWA or GatedDeltaNet-SWA. Also note that the current RAT has a clean structure without using extra short convolutions or normalization layers, which other works often adopt to boost performance.
>
> Finally, the novel chunk-based design allows partial compression of history while preserving direct access to past tokens, which helps retrieve distant information, especially in noisy contexts (see results in Reviewer 1's A2) . Unlike recent state space models that has fixed memory slots and may suffer from memory degradation on long inputs, RAT’s memory slots grows with sequence length while maintaining a fixed FLOPs reduction ratio. RAT can complements the popular local attention well by effectively and efficiently preserving long-term information. Hope these could address the reviewers’ concern.
>
> [1]. Gated Delta Networks: Improving Mamba2 with Delta Rule. ICLR’25
>
> > Q2: Because of the EMA-style construction of the "RNN" part, this also does not afford the model the capability to do state-tracking: https://arxiv.org/pdf/2404.08819.
>
> A2: The simple Eqn. (2) with per-dimension gating operates only within its own chunk (e.g., 16 or 64 tokens). Because the chunk length is so short, the simple recurrence can effectively encode the entire chunk without information loss and is therefore fully expressive for summarizing it. The cross-chunk softmax attention then handles long dependencies by providing direct access to distant tokens. This design also allows for a flexible value/state size, whereas state space models typically have fixed memory slots size even when the sequence length becomes very long.
>
> > Q3: I am surprised at the 7x training speedup and 9x generation speedup, and I am curious what the baseline comparisons are. Was flash attention used? Since I suspect the speedups gained in the paper may fade with 1) different implementation comparisons, or 2) scale of model, I think there is little gain in capability if we consider the gains from using a RAT-Attention hybrid vs a Mamba-Attention hybrid.
>
> A3: As stated in the paper (lines 185–187, bold “efficiency” in the experimental setup), we ensured fair comparisons by using FlashAttention for the attention module, associative scan for the RNN, bf16 and torch.compile for all models, and by measuring the latency of a single token mixing block (including input and output projections) on the same GH200 GPU.
>
> We analyzed the efficiency in Method Section 2.4 and showed that the observed 7×–9× speed-up is reasonable. Compared to standard attention, RAT reduces theoretical FLOPs by the chunk length (e.g., 16), and has a clean structure with no extra short convolutions or normalizations, and can be implemented very efficiently with our optimized implementation.
>   * During training, the recurrence does not involve state expansion or reduction, and different chunks can be stacked together to pass through the parallel scan. The causal masking problem in cross-chunk attention is addressed using FlexAttention and the online softmax trick.
>   * At inference time, a single-step update is used for the RNN, and the same FlashAttention can be used for cross-chunk attention.
>
> Thus, when benchmarking the latency of the temporal mixing block (including the linear projections), the observed 7×–9× speed-up is expected. We have also provided the anonymous code link in line 808 of the Appendix.
>
> For hybrid models, we interleave RAT with sliding-window attention rather than full attention for efficiency. Detailed differences and comparisons with Mamba are provided in the question above. Finally, we emphasize that RAT is a new architecture, not a hybrid or state-space model, and not an incremental variant of Mamba. We proposed the idea, explored its positional encodings, parameter allocations, and efficient implementations, and validated it across multiple tasks.
>
> > Q4: However, these efficiency gains may fade at a larger scale when MLP computation dominates.  The speedups gained in the paper may fade with 2) scale of model.
>
> A4: First, this is not specific in RAT but a general property of all temporal mixing methods: temporal mixing and channel mixing compete for runtime dominance, and which dominates depends on model dimension and sequence length. Then we analyzed the speed-up on larger models for training and generation cases, respectively.
>   * For computation-intensive training, we benchmark the latency of the temporal mixing block (including input and output projections) and find that efficiency gains decrease when the model dimension D increases because the linear projections contribute a larger fraction of FLOPs. For example, the speed-up drops from 7× to 5.5× at 100K tokens when D increases from 2048 to 4096 (Attention block: 1776.51s, RAT(L=16) block: 322.43s). However, training a 7B model typically uses tensor parallelism, which reduces the model dimension on each GPU (e.g., D=2048 with 2 GPUs), so the expected 7× speed-up can still be achieved.
>   * For memory-intensive decoding, the scaled-dot product operation in attention remains a major bottleneck, even for large models. We report latency for a single temporal mixing block with D=4096 when generating one batch of token at a specific position, and the maximum throughput of the full 7B and 13B models when generating 1024 tokens with a 3072-token prompt. RAT consistently achieves significant speed-ups over attention. On the 13B model in particular, attention utilizes GPU resources poorly due to its very limited maximum batch size, which allows RAT to achieve even greater throughput improvements.
>
> | Latency of Temporal mixing block (D=4096) | Position=4096, Batch=1024 | Position=8192, B=512 | Position=16384, B=256 | Position=32768, B=128 | Position=65536, B=64 |
> |-|-|-|-|-|-|
> | RAT(L=16)  | 3.01s| 2.7s | 2.25s | 2.21s | 2.15s  |
> | Attention  | 26.16s  | 25.82s | 25.87s | 25.8s   | 26.22s |
> | Ratio | 8.7x  | 9.6x | 11.5x| 11.7x | 12x |
>
> |Maximum throughput of a whole model | 1.3B (results in our paper) | 7B | 13B|
> |-|-|-|-|
> | RAT(L=16)| 31170 token/sec | 10103 token/sec | 5749 token/sec |
> | Attention| 3152 token/sec| 983 token/sec| 534 token/sec  |
> | Ratio  | 10.2x  | 10.3x| 10.8x |

---

> > ### Comment · Reviewer_pHG2 · 2025-08-04
> > **Response**
> >
> > > And we're not putting the EMA construction in one layer and placing the attention in the second layer.
> >
> > I did understand this. I suppose a better way to pose my question is: What is the expressivity benefit in adopting this chunking system as opposed to a mixed-layer approach?
> > In a model where linear RNNs/linear Attention and quadratic attention layers are interleaved can also (theoretically) have some of the properties (e.g. Mamba1 could reset its recurrent state) that are being touted with RAT. Is there a toy case where you can show this is not true?
> >
> >
> > Thank you for the results of comparisons against hybrid models, I have a few questions:
> > 1. Are the hybrid models' attention layers only using SWA?
> > 2. If so, is the attention context of the RAT-SWA model larger than that of the hybrid models due to the way RAT chunked attention works?

---

> ### Author Response · Authors · 2025-08-04
> **Response**
>
> Thanks for the question. As we understand, the reviewer is asking about the benefit of RAT compared to a hybrid model (e.g., using EMA in one layer and attention in another).
>   * First, their FLOPs are different. Suppose the original Transformer has O(M) FLOPs for temporal mixing operations in all layers. A mixed-layer design, even with EMA used in half the layers, still incurs at least O(M/2) FLOPs due to the remaining attention layers. In contrast, RAT(L=16) has O(M/16) FLOPs, which is another key reason why we treat and compare RAT as a standalone model rather than a hybrid.
>   * Second, in mixed-layer designs, the recurrence or linear attention layers still compress the entire sequence into fixed-size, holistic representations. This leads to memory degradation and limits fine-grained retrieval in these layers. RAT avoids this by applying recurrence only within local chunks and preserving global access via cross-chunk attention.
>   * Third, as an intermediate design between RNNs and attention, RAT offers more flexibility for hybrid modeling. For example, an EMA-attention mixed-layer design can be expressed in RAT by setting the chunk size to T in one layer and 1 in another.
>
> Regarding experiments, we only use sliding-window attention (SWA) to interleave with RAT, because SWA has fewer FLOPs than full attention and has been widely adopted in hybrid designs [1, 2, 3].
>
> We compare RAT-SWA against Mamba-SWA and GatedDeltaNet-SWA. The memory or state sizes have been listed in the results table. Each RAT layer has T/L*D memory slots.
> State space models expands the classic recurrence state to have larger memory slots.
> It can be seen that we have smaller memory/state sizes on short-context tasks (≤300 tokens, which results in only ~20 chunks for attention), and we have the same number of memory slots as Mamba2 when the sequence length is 4096 (see long-context task results in Reviewer 1 Q2A2). The memory capacity in RAT scales with sequence length while maintaining a fixed FLOPs reduction ratio.
>
> Let us know if the reviewer has further questions.
>
> T: sequence length; D: model dimension; L: chunk size
>
> [1]. Gated Delta Networks: Improving Mamba2 with Delta Rule
>
> [2]. Samba: Simple Hybrid State Space Models for Efficient Unlimited Context Language Modeling
>
> [3]. Command A: An Enterprise-Ready Large Language Model

---

> > ### Author Response · Authors · 2025-08-07
> > **Response to reviewer pHG2**
> >
> > Dear Reviewer pHG2,
> >
> > This is a resend of our earlier comment to ensure that you can access it. In our previous reply, we included “Reviewers Submitted” in the Readers list but forgot to explicitly select your reviewer ID.
> >
> > We are sending this notice again to make sure you are able to view our response. The specific reviewer ID has now been added to the Readers list in our response above.
> >
> > As the discussion period is coming to a close, could you please let us know if our responses have adequately addressed your concerns? If you have any further questions or comments, we would be happy to clarify.
> >
> > Thank you very much for your time and consideration.

---

> > > ### Comment · Reviewer_pHG2 · 2025-08-08
> > > **Thanks for the clarification**
> > >
> > > > First, their FLOPs are different. Suppose the original Transformer has O(M) FLOPs for temporal mixing operations in all layers. A mixed-layer design, even with EMA used in half the layers, still incurs at least O(M/2) FLOPs due to the remaining attention layers. In contrast, RAT(L=16) has O(M/16) FLOPs, which is another key reason why we treat and compare RAT as a standalone model rather than a hybrid.
> > >
> > > Thanks, this has made it slightly clearer for me. I do still think then that the comparison should be with a 'dilated' attention that attends to every 16th time-step, while still maintaining sliding window, though I do understand that this is hard to train.
> > >
> > > I have increased my score by 1.

---

> > > > ### Author Response · Authors · 2025-08-08
> > > > **Response to Reviewer pHG2**
> > > >
> > > > Thanks the reviewer’s understanding and consideration.
> > > >
> > > > > I do still think then that the comparison should be with a 'dilated' attention that attends to every 16th time-step, while still maintaining sliding window, though I do understand that this is hard to train.
> > > >
> > > > Thanks for pointing out your suggestion related to dilated attention. Indeed, in our experience the models using dilated attention and convolutions tend to be harder to train. As we understand it, the reviewer is suggesting a design that uses the simple recurrence in one layer, then uses a dilated attention in another layer to demonstrate the importance of our chunk-based design under the same FLOPs. We offer further explanations about efficiency and accuracy comparisons below.
> > > >
> > > > First, from an efficiency perspective, RAT has advantages over dilated attention at both training and inference, even under the same FLOPs budget. During training, dilated attention reduces parallelism by the dilation rate, since tokens within the same dilation span attend to different KV vectors. In contrast, RAT allows tokens to share the same chunk-level representation, except for their own chunk (due to causal masking and has been addressed in the paper). During inference, the sliding of the dilation pattern causes a memory issue, as all tokens’ KV cache must be stored, whereas RAT reduces the KV cache size proportionally to the chunk size.
> > > >
> > > > Second, for accuracy, we report the results on the PG19 book dataset using the 200M model reported in our paper, and conduct new experiments on the suggested design. We used a dilation rate of 64 (compared to RAT(L=128)), as they’re applied in half of the layers. Its performance is worse than a simple RNN layer. We found it has a significantly slower convergence in the beginning of training, likely due to the non-global perception of dilated attention layers. Moreover, since RAT has similar FLOPs to dilated attention but greater expressiveness from global perception, we consider it a good substitute for dilated attention.
> > > >
> > > > | Model | Perplexity on T=16384|
> > > > |-|-|
> > > > | RNN (EMA here)  | 14.52   |
> > > > | EMA + Dilated attention (dilation=64) | 15.22   |
> > > > | RAT(L=128)  | **13.44**   |
> > > >
> > > > Finally, **we think it’s important to have a single-layer design that can enable different behaviors between RNN (full-sequence compression) and attention (full-token access)**.
> > > >
> > > >   * RAT allows independent usage while addressing the major issues of each approach—RNN’s memory degradation and attention’s computational cost. Other issues can also be alleviated, such as the training parallelism of RNN (with short chunks, even non-linear RNNs become feasible) and the length generalization problem of attention (as shown in the appendix). We have demonstrated its superior performance with comparable accuracy and significantly faster processing times compared to attention.
> > > >   * RAT also supports more flexible hybrid modeling. Imagine that we can use different chunk sizes in different layers, while previous mixed-layer designs only offer the combination of two extreme cases.
> > > >
> > > > Meanwhile, regarding the reviewer’s other concerns, including the capability to do state tracking and fair efficiency and accuracy comparisons, may we ask if our responses have addressed these points adequately? As shown in our paper and rebuttal, we compared efficiency fairly under the same setup by using FlashAttention for attention, and compared accuracy with other efficient attention methods fairly by using smaller or equal value/state sizes.
> > > >
> > > > We would be happy to clarify if you have any further questions.

---

### Official Review · Reviewer_aCcd · 2025-07-02

**Clarity:** 3
**Significance:** 2
**Originality:** 2
**Rating:** 4
**Confidence:** 3

**Summary:**

The paper proposes RAT (Recurrent-Attention Transformer), an architectural design that bridges RNN efficiency and Transformer attention accuracy for language modeling. RAT partitions the input sequence into fixed-size chunks, applies a lightweight gated RNN within each chunk for capturing local dependencies, and then performs softmax attention across chunks to model long-range interactions. This design interpolates between standard RNNs and full attention depending on the chunk size, allowing a trade-off between efficiency and expressiveness.

RAT models, especially with a chunk size of 16, achieve substantial speedups—up to 9× faster generation at 4K length—while matching or surpassing attention baselines on both pretraining perplexity and downstream tasks. Moreover, a hybrid variant interleaving RAT with sliding-window attention improves both efficiency and accuracy across commonsense reasoning, long-context understanding (LongBench), and supervised fine-tuning tasks, outperforming full-attention models in several benchmarks.

**Questions:**

See weaknesses.

**Ethical Concerns:**

["NO or VERY MINOR ethics concerns only"]

**Final Justification:**

Overall, the rebuttal strengthens the work, and I am increasing the score to 4. I still consider this paper borderline. The chunking concept appears similar to the approach in the cited Transformer Quality in Linear Time (see Fig. 4). The authors should clearly articulate the differences, analyze and justify the superiority of RAT’s design, and include that work as a baseline for comparison. Another round of revision incorporating the rebuttal results and additional analysis would be needed to further strengthen the paper.

**Quality:**

2

**Strengths And Weaknesses:**

Strengths:

1. Proposes a new intermediate architecture (RAT) that blends the efficiency of RNNs with the accuracy of attention mechanisms.

2. The hybrid variant (RAT interleaved with sliding-window attention) further boosts performance, showing strong results on reasoning, summarization, and code tasks.

Weaknesses:

1. Missing strong baselines: While the paper discusses the landscape of linear and local attention methods, it only benchmarks against vanilla RNNs and standard attention. This makes the advantages of RAT less convincing, as prior work—such as hybrid architectures mixing linear and softmax attention (e.g., Hymba [1])—has shown similar trade-offs. Including such comparisons is necessary to establish RAT’s utility over existing efficient attention methods.

2. Sensitivity to chunk size: RAT’s performance declines with larger chunk sizes, requiring careful tuning to balance accuracy and efficiency, particularly for long-context or noisy inputs.

3. Lack of large-scale evaluation: The study is limited to 1.3B parameter models, leaving open questions about how well RAT scales to larger models used in practical deployment.

[1] Hymba: A Hybrid-head Architecture for Small Language Models

---

> ### Author Rebuttal · Authors · 2025-07-31
>
> We thank the reviewer for the feedback and suggestions. Here are the responses.
> > Q1: Missing strong baselines: While the paper discusses the landscape of linear and local attention methods, it only benchmarks against vanilla RNNs and standard attention. This makes the advantages of RAT less convincing, as prior work—such as hybrid architectures mixing linear and softmax attention (e.g., Hymba [1])—has shown similar trade-offs. Including such comparisons is necessary to establish RAT’s utility over existing efficient attention methods.
>
> A1: To address the reviewer’s concerns, we clarify RAT's relationship to hybrid models and provide comparisons to efficient attention methods. RAT uses a simple recurrence within each chunk and applies softmax-based attention across chunks.
>
>   * Thus, RAT is not a hybrid model. It applies the same mechanism across all heads, tokens, and layers, and over time, while hybrid models like Hymba alternate state-space modules and attention in different heads and Griffin [1] alternates them in different layers. The hybrid strategy is orthogonal to RAT. In this paper, we explored interleaving RAT with sliding-window attention (RAT-SWA). Future work could adopt Hymba-like strategies, such as using RAT in some heads and attention in others, or applying RAT with different chunk sizes in different heads.
>
>   * Second, we mainly compared RAT to vanilla RNNs and standard attention because RAT directly interpolates between these two models and even achieves comparable or better performance than the strong baseline attention. Although RAT is also not a linear attention model (the simple recurrence uses per-dimension gating rather than per-head gating, the softmax-based attention is used across chunks), we noted in the paper (lines 245–246 with footnotes) that our best model outperforms strong baselines reported in a recent linear attention work [2], where the models below were trained on the same datasets, model sizes, and sequence lengths as ours. For completeness, we post detailed results below and will add them in the revision.
> | Model                   | value/state size (T=300, D=2048, N=24) | arc_challenge | arc_easy     | hellaswag    | lambada | piqa         | winogrande | boolq |
> |-|-|-|-|-|-|-|-|-|
> |    |     | acc/acc_norm  | acc/acc_norm | acc/acc_norm | acc     | acc/acc_norm | acc        | acc   |
> | Mamba1   | 64ND        | -/35.40       | 69.52/-      | -/52.91      | 43.98 | 71.32/-      | 52.95      | 61.13 |
> | Mamba2  | 256ND    | -/37.88       | 72.47/-      | -/55.67      | 45.66 | 71.87/-      | 55.24      | 60.13 |
> | DeltaNet  | 128ND    | -/35.66       | 68.47/-      | -/50.93      | 42.46 | 70.72/-      | 53.35      | 55.29 |
> | GatedDeltaNet    | 288ND  | -/38.39       | 71.21/-      | -/55.76      |46.65 | **72.25**/-      | 57.45      | 60.24 |
> | RAT(L=16)  | ~20ND   | 36.35/**39.33**   | **72.64**/67.80  | 43.27/**56.44**  | 44.4    | 72.03/72.69  | 53.2       | **63.3**  |
> | **Hybrid**     |          |         |    |      |         |     |      |       |
> | GatedDeltaNet-SWA(2048) | 288N/2D-300N/2D      | -/40.10       | 71.75/-      | -/56.53      | 47.73   | 72.57/-      | **58.4**       | 63.21 |
> | RAT(L=16)-SWA(1024)     | ~20N/2D-300N/2D     | 37.20/**40.78**   | **72.56**/68.22  | 44.33/**57.85**  | **49.33**   | **73.29**/73.94  | 56.91      | **63.21** |
>
>
>     These commonsense reasoning tasks have very short contexts (≤300 tokens), so RAT with L = 16 at most has 20ND value size. Even so, it surpasses other models in 4 of 7 tasks, and RAT-SWA outperforms GatedDeltaNet-SWA in 6 of 7 tasks. Note that the current RAT has a clean structure without using extra short convolutions or normalization layers, which other works often adopt to boost performance.
>
> Hope these can address the reviewer’s concern.
>
> [1]. Griffin: Mixing Gated Linear Recurrences with Local Attention for Efficient Language Models
>
> [2]. Gated Delta Networks: Improving Mamba2 with Delta Rule. ICLR’25
> > Q2: Sensitivity to chunk size: RAT’s performance declines with larger chunk sizes, requiring careful tuning to balance accuracy and efficiency, particularly for long-context or noisy inputs.
>
> A2: Thanks for the question. In this paper, we showed that using a short chunk size with the simple recurrence is both performant and highly efficient. The chunk design in RAT can also help with long-context and noisy-input cases compared to recent state-space and linear attention models. We’ll elaborate on each point below.
>
>   * First, it is natural that performance declines with larger chunk sizes, as FLOPs also decrease. Other models have similar trade-offs (e.g., the state expansion factor in state-space models). In the paper, we showed that L = 16 yields 7×/9× speed-ups while achieving comparable results with the strong attention baseline on challenging tasks, including LongBench (Table 3), SFT (Table 4), and the Ruler benchmark (Table 12 in Appendix).
>   * Second, when the context becomes very long, the number of chunks (memory slots) in RAT grows with sequence length. This helps maintain accuracy while preserving the same FLOPs reduction ratio, while state-space or linear attention models, which has fixed memory slots, can suffer from memory degradation on very long inputs. For noisy inputs, the cross-chunk softmax attention in RAT directly retrieves past information rather than compressing all history together (as in Mamba2), which we believe is advantageous. We report results on the retrieval-heavy tasks Needle-in-Haystack tasks and QA tasks in LongBench, which contain very noisy backgrounds. It can be seen that RAT outperforms others in most tasks.
>
> | Model          | Value/state size(T=4096, D=2048, N=24 (layer)) | NIAH-1 | NIAH-2 | NIAH-3 |
> |-|--|-|-|--|
> | DeltaNet       | 128ND      | **99**     | 18.6   | 22.4   |
> | Mamba2         | 256ND| 65.4   | 56.2   | 4.6    |
> | Gated DeltaNet | 288ND  | 91.4   | 92.2   | 27.6   |
> | RAT(L=16)| 256ND   | 97     | **97.6**   | **56.4**   |
>
> | Model  | Value/state size (T=4096, D=2048, N=24) | NQA  | Qasper | MQA  | HQA  | WQA  | MSQ |
> |-|-|-|-|-|-|-|-|
> | Mamba2  | 256ND| 11.1 | 11.3   | 18.6 | 11.8 | 15.1 | **6.7** |
> | DeltaNet | 128ND  | 12.9 | 10.8   | 21.5 | 10.9 | 13.2 | 5.1 |
> | GatedDeltaNet| 288ND | 14.1 | 14     | 23.3 | 13.7 | 14.4 | 5.8 |
> | RAT(L=16)  | 256ND  |**14.5**| **16.1**   | **28.3** | **16.7** | **18.9** | 6.3 |
>
> Therefore, we think that the chunk design proposed in RAT is meaningful and important. Currently, with the simple recurrence, performance can drop at very large chunk sizes as the model approaches vanilla RNN behavior. Future work may explore more advanced RNNs for different chunk sizes, such as using state-expanded recurrence for large chunks and nonlinear RNNs for short chunks when computational efficiency is no longer an issue.
>
> > Q3: Lack of large-scale evaluation: The study is limited to 1.3B parameter models, leaving open questions about how well RAT scales to larger models used in practical deployment.
>
> A3: Thanks for pointing this out, and we have highlighted it as one of the limitations in our paper. Since we are introducing a novel architecture in this work, we prioritized model and architecture ablations (including the core idea, parameter allocation, positional encoding, and efficient implementation) and benchmarked RAT across multiple tasks. We believe these provide more insight into how the new architecture works, especially given the limited compute resource available to us.
>
> Although we did not train 7B or 13B models in this paper, we provide speed-related results on larger models below to demonstrate its efficiency potential at scale. We also noted in the paper that RAT can be easily combined with tensor parallelism (as the core part is head-independent) and context parallelism, making the method scalable to larger settings. With reduced number of key and value, it might even avoid the need for ring attention in context parallelism and thus further saves the communication cost.
>
> The first table below tested the generation latency with 7B model dimension 4096 at a specific position, yielding 8-12x speed-up. The second table shows the maximum throughput of the whole model by generating 1024 tokens given the prompt of 3072. In particular, on the 13B model, attention utilizes GPU resources poorly due to the very limited maximum batch size, allowing our method to achieve even greater improvements in terms of maximum throughput.
> | Latency of Temporal mixing block (D=4096) | Position=4096, Batch=1024 | Position=8192, B=512 | Position=16384, B=256 | Position=32768, B=128 | Position=65536, B=64 |
> |-|-|-|-|-|-|
> | RAT(L=16)   | 3.01s| 2.7s | 2.25s | 2.21s  | 2.15s  |
> | Attention   | 26.16s  | 25.82s | 25.87s  | 25.8s   | 26.22s |
> | Ratio | 8.7x  | 9.6x | 11.5x| 11.7x   | 12x |
>
> | Maximum throughput of a whole model | 1.3B (results in our paper) | 7B              | 13B            |
> |-------------------------------------|-----------------------------|-----------------|----------------|
> | RAT(L=16)                           | 31170 token/sec             | 10103 token/sec | 5749 token/sec |
> | Attention                           | 3152 token/sec              | 983 token/sec   | 534 token/sec  |
> | Ratio                               | 10.2x                       | 10.3x           | 10.8x          |

---

> > ### Author Response · Authors · 2025-08-06
> > **Discussion phase closing soon**
> >
> > Dear Reviewer aCcd,
> >
> > Thank you again for your constructive feedback. We sincerely appreciate the time you have taken to provide valuable feedback for our work. As we are getting closer to the end of the discussion period, could you please let us know if our responses above have adequately addressed your concerns? We remain at your disposal for any further questions.

---

> > ### Comment · Reviewer_aCcd · 2025-08-07
> > **Thank you for the rebuttal**
> >
> > Thank you for the detailed rebuttal and additional results. The new comparisons to strong baselines (e.g., hybrid models) and the clarification on RAT’s design help address some concerns. The chunk-size sensitivity explanation and LongBench results are useful. The added large-scale latency and throughput results are appreciated, but the evaluation remains limited in scope—particularly regarding accuracy, real-world deployment efficiency, and accuracy–efficiency trade-offs under different settings, especially when considering related system optimization like FlashAttention for standard attention. Overall, the rebuttal strengthens the work, while more comprehensive benchmarking (with not only attention but also linear attention) is needed to fully validate the superiority. The reviewer acknowledges that this may require significant computational resources, which could be a bottleneck.
> >
> > Additionally, the chunking concept appears similar to the approach in the cited "Transformer Quality in Linear Time" (see Fig. 4). If there are substantial differences, they should be clearly articulated, and that work should also be included as a baseline for comparison.

---

> ### Author Response · Authors · 2025-08-07
> **Response to reviewer aCcd**
>
> Thanks for the reply. We’d like to further explain some points that might have been overlooked or misunderstood by the reviewer.
> > but the evaluation remains limited in scope—particularly regarding accuracy, real-world deployment efficiency, and accuracy–efficiency trade-offs under different settings, especially when considering related system optimization like FlashAttention for standard attention.
>
> We acknowledge that 7B/13B model training is valuable. However, given RAT is a novel architecture, we focused our initial efforts on the 1B scale to allow more controlled and interpretable insights within our computational constraints. Regarding real-world deployment efficiency, **RAT’s efficiency is compared with FlashAttention as stated in the efficiency setup of the paper. The efficiency comparison is fair.**
>
> > while more comprehensive benchmarking (with not only attention but also linear attention) is needed to fully validate the superiority
>
> In the rebuttal, **we have provided comparisons with linear attention models including Mamba2, DeltaNet, and GatedDeltaNet**. While using value/state sizes that are smaller or equal, we showed RAT’s superiority over them. With the per-head gating mechanism, recent state space models are indeed linear attention as well.
>
> > Additionally, the chunking concept appears similar to the approach in the cited "Transformer Quality in Linear Time" (see Fig. 4). If there are substantial differences, they should be clearly articulated, and that work should also be included as a baseline for comparison.
>
> Thanks for pointing this out. We cited that work in the paper and described its use of local attention. Its design is very different from ours, and we will further clarify this in the revision. The GAU model applies softmax-based attention within local chunks and uses linear attention across chunks to cache history. The two outputs are simply added together, whereas we use recurrence within chunks and softmax-based attention across chunks, organizing the outputs in a hierarchical manner.
>
> As for the comparison, we did not include it for the following two reasons, corresponding to its two components: linear attention and local attention.
>   * In our early experiments on WikiText-103 with 4-layer models, we found that it performs worse than Mamba2 (e.g., 31.47 ppl for Mamba2 vs. 32.41 ppl for GAU). Therefore, we believe our superiority over Mamba2 already implies superiority over GAU.
>   * Regarding its use of local attention, we consider local attention and our method to be complementary. We have provided hybrid model results that interleave RAT with sliding window attention as a starting point. We see the integration of more complex local attention designs (e.g., those with additional components like linear recurrence) with RAT as future work.
>
> Let us know if the reviewer has further questions.

---

> > ### Comment · Reviewer_aCcd · 2025-08-09
> > **Thank you for the follow-up**
> >
> > Thank you for the further explanation. The differences from the cited FLASH work indeed need clarification in paper. While both approaches share a similar chunked framework, your method differs in both inter-chunk and intra-chunk processing. Additional analysis and ablation studies would help readers better understand what works best and why the RAT design is most suitable for inter- and intra-chunk handling.

---

> ### Author Response · Authors · 2025-08-09
> **Thank you for the suggestion**
>
> We thank the reviewer for the constructive suggestion and for further explaining the concerns.
>
> > Thank you for the further explanation. The differences from the cited FLASH work indeed need clarification in paper. While both approaches share a similar chunked framework, your method differs in both inter-chunk and intra-chunk processing. Additional analysis and ablation studies would help readers better understand what works best and why the RAT design is most suitable for inter- and intra-chunk handling.
>
> Though we both use the concept of chunking, **we still think our work and [1] adopt fundamentally different chunking frameworks—not only because of the different components used, but also because of how the two outputs are combined**: [1] simply adds the outputs of inter-chunk and intra-chunk computations, whereas RAT organizes them hierarchically, with recurrence applied over the key and value before the inter-chunk attention. We now provides analyses and ablations below.
>
> As we had stated, we explored their method in our early experiments and found that it does not outperform Mamba2. In the appendix C.1 and Table 9, we also explored a reversed design of RAT (local attention within chunks + recurrence across chunks) and found it performs worse than RAT, because the local attention design does not utilize the FLOPs as effectively as cross-chunk attention.
>
> To provide more comprehensive understanding, we take the reviewer’s suggestion to include [1] and will update our Table 9 accordingly. The first two rows below were conducted and reported in our paper. All experiments are conducted on a Book dataset with 200M-parameter models, and we controlled FLOPs to be the same across methods.
>
> |                       | Intra-chunk     | Inter-chunk      | Way to organize them | FLOPs   | PPL on T=16384 |
> | --------------------- | --------------- | ---------------- | -------------------- | ------- | -------------- |
> | RAT(L=64)             | Recurrence      | Attention  | Hierarchically       | O(T/64) | **13.34**          |
> | RAT-Reversed (C=64) | Attention | Recurrence       | Hierarchically       | O(T/64) | 14.06          |
> | Method in [1] (C=128) | Attention | Linear attention | Add together         | O(T/64) | 14.4           |
>
> We think the weaker results of the method in [1] can be attributed to two main factors:
>   * First, as analyzed in Table 9 and Appendix C.1, RAT utilizes FLOPs more effectively than methods with local attention by applying recurrence to simpler local regions, while inter-chunk attention prevents memory degradation. In contrast, under the same FLOPs budget, local attention cannot cover a large context, and long-range information must rely on memory-degrading components such as recurrence and linear attention. In particular, [1] uses linear attention without decay, and in our early experiments, upgrading it to more advanced linear attention variants made the next issue more severe.
>
>   * Second, as also observed in our early experiments, their approach of directly adding the outputs of softmax-based attention and linear attention degraded performance. We think this may be due to differences in output scale and potential representational conflicts when adding them directly.
>
>
> T: sequence length; C: number of chunks; L: chunk size
>
> [1]. Transformer Quality in Linear Time

---

### Official Review · Reviewer_CiSa · 2025-07-12

**Clarity:** 3
**Significance:** 4
**Originality:** 4
**Rating:** 5
**Confidence:** 5

**Summary:**

This paper introduces RAT (Recurrence-Attention Transition), a novel architecture that interpolates between the efficiency of RNNs and the expressiveness of attention by chunking sequences.
Within each chunk (size L,16 in the paper), a lightweight gated linear RNN captures local dependencies, while softmax attention spans chunks to model long-range interactions.
This design reduce the burdens of the quadratic complexity of full self-attention (O(T²)) by reducing inter-chunk attention to O(C·D) flops, where C = T/L is the number of chunks. RAT flexibly trades off efficiency and accuracy via L: L=1 mimics attention, while L=T collapses to an RNN.

Experiments on 1.3B-parameter models demonstrate 7× faster training (1M-token sequences) and 9× faster generation (4K-length) with L=16, while maintaining perplexity within 0.06 of full attention.
On short-context benchmarks (e.g., HellaSwag, PIQA), RAT w/ L 16 trails attention by <2 points but improves with hybrid RAT-SWA (sliding-window attention + RAT), gaining +1 point on commonsense tasks. On long-context LongBench, RAT-SWA outperforms attention on 12/14 tasks, including +4 points on code completion.

**Questions:**

* I'm curious if the author try other recent RNN variants in RAT, for example, HGRN and RWKV4?
* In terms of param allocation, if I understand correctly, the final layer would have 6d^2 params with shared q/k? Did the authors tied key and forget gate param allocation as in HGDN2?
* It would make the paper more comprehensive if the authors add some discussions with GAU in terms of local/global chunk design, and Log Linear Attention in terms of O(log N) caching. Some recent works, e.g., GSA, also makes some efforts in combing attention with RNNs
* I'm especially curious what if combining RAT with some state expansion techniques like in Mamba or Hedgehog, which intuitively could make the 1d RNNs more powerful once expanded to 16x or larger state size

HGRN: https://arxiv.org/abs/2311.04823

GAU: https://arxiv.org/pdf/2202.10447

GSA: https://arxiv.org/abs/2409.07146

Hedgehog: https://arxiv.org/abs/2402.04347

**Ethical Concerns:**

["NO or VERY MINOR ethics concerns only"]

**Limitations:**

yes

**Quality:**

3

**Strengths And Weaknesses:**

I find the RAT (Recurrence-Attention Transition) architecture to be an elegant and natural solution for balancing expressiveness and efficiency in long-context modeling. Unlike recent works (e.g., Log-Linear Attention), which impose hierarchical inductive biases with O(log N) memory growth, RAT’s chunking strategy feels more intuitively grounded:
* At 1M context, RAT with L=16 retains a fixed 64K KV cache, a constant-sized compression that scales gracefully without the logarithmic overhead of alternatives (log₂(1M)/log₂(4K) ≈ 2× growth in Log-Linear Attention).
* Hybridization with SWA further reinforces its practicality: local attention handles short-range dependencies (where RNNs might underperform), while RAT’s inter-chunk attention preserves global coherence. This synergy outperforms pure attention or SSMs on LongBench, suggesting the architecture generalizes well beyond synthetic benchmarks.

And I am particularly eager to see the results of RAT in terms of param and context scaling (if possible):
* will RAT’s efficiency gains persist under larger model regimes like 7B/14B?
* how does RAT fare against other RNNs like Mamba-2/HGRN2/DeltaNet/GDN on retrieval-heavy tasks?

---

> ### Author Rebuttal · Authors · 2025-07-31
>
> We would like to thank the reviewer for their constructive feedback. Following is our responses.
> > Q1: will RAT’s efficiency gains persist under larger model regimes like 7B/14B?
>
> A1: In response to the reviewer’s question we provide a detailed efficiency study which we will add to the appendix of our paper:
>
>   * First, as the model dimension D grows, the latency of the temporal mixing block is increasingly dominated by the input and output linear projections, so the efficiency gains during compute-intensive training diminish. For instance, when increasing D from 2048 to 4096, the speed-up drops from 7× to 5.5× for 100K tokens (Attention block: 1776.51s vs. RAT(L=16) block: 322.43s). However, training a 7B model usually involves tensor parallelism, which effectively reduces the dimension per GPU (e.g., D=2048 with 2 GPUs), allowing the expected speed-up to still be achieved.
>
>   * Second, the scaled dot-product operation of attention remains a major bottleneck in memory-intensive decoding, even for large models and especially in the long–context settings. We report the latency of a single temporal mixing block with D = 4096 for generating a batch of tokens at a specific position and the maximum throughput of the full 7B and 13B models for generating 1024 tokens with a 3072-token prompt. RAT shows a significant speed-up over attention. In particular, on the 13B model, the maximum batch size of attention is very limited, leading to poor GPU utilization and allowing RAT to achieve even greater throughput gains.
>
> | Latency of Temporal mixing block (D=4096) | Position=4096, Batch=1024 | Position=8192, B=512 | Position=16384, B=256 | Position=32768, B=128 | Position=65536, B=64 |
> |-|-|-|-|-|-|
> | RAT(L=16)   | 3.01s| 2.7s | 2.25s | 2.21s  | 2.15s  |
> | Attention   | 26.16s  | 25.82s | 25.87s  | 25.8s   | 26.22s |
> | Ratio | 8.7x  | 9.6x | 11.5x| 11.7x   | 12x |
>
> | Maximum throughput of a whole model | 1.3B (results in our paper) | 7B              | 13B            |
> |-|-|-|-|
> | RAT(L=16) | 31170 token/sec | 10103 token/sec | 5749 token/sec |
> | Attention | 3152 token/sec  | 983 token/sec   | 534 token/sec  |
> | Ratio  | 10.2x     | 10.3x           | 10.8x          |
>
> > Q2: how does RAT fare against other RNNs like Mamba-2/HGRN2/DeltaNet/GDN on retrieval-heavy tasks?
>
> A2: Thanks for this suggestion. We present comparisons on the Ruler Benchmark, which contains challenging synthetic tasks for retrieval ability (NIAH tasks), as well as QA tasks in LongBench. We compare our model against the results from the recent paper [1], which trained several linear attention models on the same dataset, with the same model size and sequence length as ours. As shown in the two tables below, we achieve better performance with the same or even smaller value/state size. We think the improvement comes from RAT’s direct access to the past, which can be beneficial in noisy backgrounds compared to other models that compress all history together.
>
> |Model  | value/state size (T=4096, D=2048, N=24) | NIAH-1 | NIAH-2 | NIAH-3 |
> |-|-|--------|--------|--------|
> | DeltaNet       | 128ND  | **99**     | 18.6   | 22.4   |
> | Mamba2         | 256ND  | 65.4   | 56.2   | 4.6    |
> | Gated DeltaNet | 288ND                                      | 91.4   | 92.2   | 27.6   |
> | RAT(L=16)      | 256ND                                      | 97     | **97.6**   | **56.4**   |
>
> | Model         | value/state size (T=4096, D=2048, N=24) | NQA  | Qasper | MQA  | HQA  | WQA  | MSQ |
> |---------------|-----------------------------------------|------|--------|------|------|------|-----|
> | Mamba2        | 256ND   | 11.1 | 11.3   | 18.6 | 11.8 | 15.1 | **6.7** |
> | DeltaNet      | 128ND    | 12.9 | 10.8   | 21.5 | 10.9 | 13.2 | 5.1 |
> | GatedDeltaNet | 288ND    | 14.1 | 14     | 23.3 | 13.7 | 14.4 | 5.8 |
> | RAT(L=16)     | 256ND    | **14.5** | **16.1**   | **28.3** | **16.7** | **18.9** | 6.3 |
>
> Specifically, the value/state size calculation is based on sequence length T, model dimension D, and number of layers N. RAT (L=16) has 4096/16ND = 256ND value vectors. For other models, we multiply the value dimension with the state expansion factor. Note that Mamba2 and GatedDeltaNet expand the value dimension compared to standard attention. Finally, for the Ruler Benchmark (NIAH tasks), we would like to mention that in Appendix C.4 and Table 12, we compared RAT with attention in terms of retrieval ability and obtained comparable performance after light fine-tuning for prompt adaptation. Here, we directly evaluate RAT’s performance in the same way as theirs to ensure a fair comparison.
>
> [1]. Gated Delta Networks: Improving Mamba2 with Delta Rule. ICLR’25
> > Q3: I'm curious if the author try other recent RNN variants in RAT, for example, HGRN and RWKV4? I'm especially curious what if combining RAT with some state expansion techniques like in Mamba or Hedgehog, which intuitively could make the 1d RNNs more powerful once expanded to 16x or larger state size.
>
> A3: We tried this in our early experiments with a design inspired from  Mamba2 architecture, but found that using state expansion and reduction in the intra-chunk RNN, while enhancing expressiveness, introduced additional optimization difficulties. The experiments were conducted on a 4-layer model on WikiText-103 with model dimension 256, sequence length 256, and chunk size 64. These issues might disappear in larger models and datasets. And the added expressivity could become more important for very long chunks (e.g., 2k or 4k), but as an initial step, we primarily focus on shorter chunks and leverage softmax attention more, which currently dominates modern architectures. In detail, we considered two cases:
>
>   * State expansion → cross-chunk attention → state reduction: We applied state expansion on the “key” and “value” vectors and then performed softmax-based attention on the expanded states across chunks, followed by state reduction. We observed that the loss curve flattened early and remained at a higher loss level than its 2-D RNN counterparts. We think this is because involving many matrix multiplications without intermediate non-linearity can make optimization more challenging.
>   * State expansion → reduction → cross-chunk attention: Here, we applied expansion and reduction within each chunk and then performed inter-chunk attention. This approach only performs well when the state size is relatively small (e.g., 4 or 16). Given 64 as the chunk size, it is possible that the extra expressiveness of even larger state size is unnecessary and instead makes optimization harder.
>
> Under the scope of RAT, 2-D linear recurrence could be explored  in the future for large chunk-length cases.  We also mentioned in the paper that non-linear RNNs could be explored for short chunk-length cases because the computational efficiency is no longer an issue there. For now, to introduce the idea, we use the simple and fast 1-D linear recurrence in the paper.
>
> > Q4: In terms of param allocation, if I understand correctly, the final layer would have 6d^2 params with shared q/k? Did the authors tied key and forget gate param allocation as in HGDN2?
>
> A4: The final layer is still 4d^2 parameters, and we did not tie the key and forget gates. We share “query” and “key” across different heads. And due to the existence of the per-dimension forget gate over key, such a design won’t collapse into a single-head attention. Thus, there’s the d^2 parameters for forget gate, output gate, value vector, and output projection, respectively. Query and key vectors here only require a small portion of parameters. It is D*256 in our 1.3B model with D=2048.
>
> > Q5: It would make the paper more comprehensive if the authors add some discussions with GAU in terms of local/global chunk design, and Log Linear Attention in terms of O(log N) caching. Some recent works, e.g., GSA, also makes some efforts in combining attention with RNNs.
>
> A5: Thanks for the suggestions. Here are some discussions and we’ll add them in the revision.
>   * Yes, we cited GAU in the paper and described its use of local attention. In the revision, we will further clarify that GAU applies softmax-based attention within local chunks and uses linear attention across chunks to cache history. The two outputs are simply added together, whereas we use recurrence within chunks and softmax-based attention across chunks and organize their outputs in a hierarchical manner.
>
>   * The concurrent work Log-Linear attention focuses on the fixed state size limitation of linear attention by increasing the memory cache size in a logarithmic manner. Our work also targets the fixed memory size issue in recurrent models, which we address by applying softmax-based attention sparsely to enable direct access to distant tokens.
>
>   * As for the interesting paper GSA, sorry for missing it. GSA aggregates and caches all “key” and “value” vectors into a 2D memory matrix, where the “query” attends to the compressed key and value representations. Compared to standard attention, this approach can be implemented as a two-pass linear attention method and is thus more efficient. A key difference from our method is that we do not aggregate the entire sequence’s keys and values; instead, we store each chunk’s information through a simple recurrence and use cross-chunk softmax-based attention to directly retrieve history. This design enables a flexible state/memory size depending on sequence length and allows for efficient implementation via FlexAttention.

---

> > ### Comment · Reviewer_CiSa · 2025-08-02
> >
> > Thank you for your thoughtful response. I have reviewed your comments as well as those from the other reviewers. This is a strong paper, and I look forward to seeing the improved version.

---

> > > ### Author Response · Authors · 2025-08-06
> > > **Comment**
> > >
> > > Dear reviewer CiSa,
> > >
> > > Thank you once again for the encouraging feedback and helpful suggestions to improve the paper. We sincerely appreciate the time you have taken to provide valuable feedback for our work. We'll improve our paper based on your thoughtful suggestions.
> > >
> > > Sincerely, The Authors

---

### Note · Authors · 2025-08-13

We highlight our paper’s main contributions and concerns in the rebuttal. Our main contributions are:
  * We notice that RNNs compress the full sequence into a fixed-size holistic representation, causing memory degradation and limiting fine-grained retrieval in long contexts, while attention offers high accuracy via full-token access but at high computational cost. To combine the strengths of both, we propose a novel intermediate layer called RAT, a chunk-based design that compresses only local context via recurrence while preserving global access through cross-chunk attention. It mitigates memory degradation, enables direct access to past tokens, and maintains high efficiency.
  * RAT is simple, scalable, and efficient. We explored design choices, efficient implementation, and discussed its parallelism compatibility. Importantly, it is not a hybrid model but is orthogonal to hybrid methods. We also evaluated interleaving RAT with sliding-window attention (SWA) and found them complementary.
  * We validate RAT on 1.3B scale on diverse tasks, including short-context reasoning (Table 2), LongBench (Table 3), SFT (Table 4), and Ruler (Table 12, appendix). RAT and its SWA hybrid match strong attention baselines in accuracy while achieving much higher speed and lower KV cache usage (e.g., 9× decoding speed-up for a RAT block, 10× max throughput for the full model).

Main concerns in the rebuttal:
  * Performance vs. linear attention/state-space models: As noted in the paper (line 244-246 with footnotes), our best model surpasses the best model in a recent work that trained many linear attention models under the same setting as ours. We did not include their results in the table to keep the comparison with strong attention direct. However, given the reviewers’ concern, we listed them in the rebuttal and will add them in the revision.
  * Efficiency on 7B/13B models: We added such results. All our efficiency results follow a fair setup such as FlashAttention for attention, with details in sections 2.4 and 3.1, and an anonymous code link in the appendix.
  * Ablation studies: In the last two days of the discussion phase, the second and third reviewers suggested interesting ablations. While we had included key ones in Figure 4(a) (design choice) and Table 9 (reversed design with local attention), we added comparisons to FLASH and (sliding) dilated attention. The latter is less efficient than RAT in training parallelism and KV cache reduction, and harder to train.

---

### Decision · Program_Chairs · 2025-09-17

**Decision:**

Accept (poster)

**Comment:**

The paper proposes RAT, a chunked recurrence-plus-attention mechanism that achieves strong efficiency gains (up to 9–10× speedup) while maintaining or improving accuracy on both short- and long-context benchmarks. The approach is simple and empirically validated at the scale of1.3B, with further evidence provided in the rebuttal for 7B/13B settings. The main concerns raised were incomplete benchmarking against other efficient or hybrid attention methods and the need for clearer distinctions from related chunked/state-space approaches. While one reviewer remained unconvinced, the additional results and analysis alleviated many concerns, and the overall consensus is that this work makes a meaningful and novel contribution. I recommend acceptance, with the expectation that the camera-ready version will resolve the rest of the concerns.